# Recent Advances in the Phytochemistry of Bryophytes: Distribution, Structures and Biological Activity of Bibenzyl and Bisbibenzyl Compounds

**DOI:** 10.3390/plants12244173

**Published:** 2023-12-15

**Authors:** Kakali Sen, Mohammad Imtiyaj Khan, Raja Paul, Utsha Ghoshal, Yoshinori Asakawa

**Affiliations:** 1Department of Botany, University of Kalyani, Kalyani 741245, Indiaghoshalsamratutsha@gmail.com (U.G.); 2Department of Biotechnology, Gauhati University, Guwahati 781014, India; imtiyaj@gauhati.ac.in; 3Institute of Pharmacognosy, Tokushima Bunri University, Tokushima 770-8514, Japan; asakawa@ph.bunri-u.ac.jp

**Keywords:** bryophytes, bibenzyls, bisbibenzyls, extraction, characterization, structure–activity relationship

## Abstract

Research on bryophyte phytochemistry has revealed the presence of different phytochemicals like fatty acids, terpenoids, small phenolic molecules, etc. Small phenolic molecules, i.e., bibenzyls (of two aromatic rings) and bisbibenzyls (four aromatic rings), are unique signature molecules of liverworts. The first bisbibenzyls marchantin A and riccardin A were discovered in two consecutive years, i.e., 1982 and 1983, respectively, by Asakawa and coworkers. Since then, about 70 bisbibenzyls have been reported. These molecules are characterized and identified using different spectroscopic techniques and surveyed for different bioactivity and structure–activity relations. Biochemistry is determined by the season, geography, and environment. In this review, quantitative and qualitative information on bibenzyls and bisbibenzyl compounds and their distribution in different liverworts across, geographies along withtraditional to advanced extraction methods, and characterization techniques are summarized. Also, a comprehensive account of characteristic spectra of different bisbibenzyl compounds, their subtypes, and their basic skeleton patterns are compared. A comprehensive table is provided here for the first time presenting the quantity of bibenzyls, bisbenzyls, and their derivatives found in bryophytes, mentioning the spectroscopic data and mass profiles of the compounds. The significance of these compounds in different bioactivities like antibiotic, antioxidative, antitumor, antivenomous, anti-influenza, insect antifeedant, cytotoxic, and anticancerous activities are surveyed and critically enumerated.

## 1. Introduction

Bryophytes are non-vascular land plants belonging to the non-tracheophyte group that do not produce bona fide lignin polymers even though many of the genes for its biosynthesis are present [1,2,3]. There are three bryophyte lineages: Anthocerotophyta (hornworts, consisting of ~250 species), Marchantiophyta (liverworts, ~9000 species), and Bryophyta (mosses, ~12,700 species) [4]. Bryophytes, particularly Marchantiophyta/liverworts, emerged ~500 million years ago on Earth [5], and they are the ancestors of all land plants [1,6]. Liverworts and mosses form a monophyletic clade [7], whereas hornworts are sisters to them [3].

Bryophytes started blooming under challenging environmental conditions of high-intensity solar radiation and high CO_2_ level even though they were biochemically constrained for photosynthesis because of lower Rubisco expression and anatomical limitations of CO_2_ diffusion [8]. Another challenge they faced was adapting their life in a water-deficient and UV-abundant terrestrial environment; see the review by [9,10]. Bryophytes are morphologically inconspicuous, meaning they lack well-developed tissue structures to protect them from stress factors. However, to make up for their morphological shortcomings, bryophytes have evolved to synthesize specialized phytochemicals to contribute to their ecometabolics [10], which is the metabolic response to ecological challenges through the production of specific compounds [9]. Among the array of metabolites produced by bryophytes, flavonoids protect from light [11], sesquiterpenoids and flavonoids act counter biotic stress [11,12], most glycosides act as shields against low temperature and desiccation [13], and methoxyphenols and cinnamic acid derivatives constitute anatomical structures that support growth and development [14]. The presence of bibenzyls (compounds with two aromatic rings) and bisbibenzyls (compounds with four aromatic rings), a unique group of phenolic molecules of liverworts present in oil bodies, are abundant in bryophytes, with a structural diversity. Mosses are generally devoid of these molecules, as oil bodies are absent in them [9], and the role of these oil bodies is known to prevent desiccation.

Metabolites of these minuscule organisms perform the role of overcoming the challenges they face in nature [9,14,15,16,17]. Therefore, when they are extracted separately and tested, purified andtested, they show many bioactive properties, including antibacterial, antifungal, antioxidative, antivenomous, anticancerous, anti-inflammatory, antiulcer and others [18,19,20] of human interest. As a diverse array of metabolites and their function has been reported, for convenience, we restricted our major focus to bibenzyl and bisbibenzyl compounds of liverworts and other bryophytic taxa as reported in the literature. A number of reviews have covered the bioactivity and structure–activity relation of bisbibenzyls [21,22,23,24], the ethnobryological uses of bryophytes [25,26,27], and the synthesis and structural diversity of said compounds [15,28,29]. The gap in the previous reviews was the absence of comprehensive quantitative information on specialized bryophyte metabolites for ready reference and the mechanistic explanation. Also, the difference in extraction protocols vis-a-vis the corresponding yield has not been critically appraised. As such, in this review, we complement the literature with recent updates of the liverwort taxa surveyed for the extraction and characterization of signature molecules of liverworts, i.e., bibenzyls and bisbibenzyls, and their quantities in different taxa, the conventional extraction method vis-à-vis recent micromanipulation techniques for cell-based characterization, and the structure–activity relationship. We critically analyze the knowledge gap in the above-mentioned areas to lay the foundation for future research questions.

## 2. Extraction and Purification of Major Bioactive Compounds

Different extraction methods involving different solvent systems have been used for the extraction of bryophyte phytoconstituents, like macrocyclic phenolic compounds, such as bibenzyls and bisbibenzyls. To increase the yield, the extraction process is complemented by treatments like sonication or an extended period of extraction, for example, a week. Depending upon the chemical nature/polarity, methanol (MeOH), ethanol (EtOH), diethylether (Et_2_O), and other solvents have been used for preparing crude extracts. The fractionation of the extract or the chromatographic separation of various compounds in the extract through adsorption or gel permeation is carried out using different mixtures of solvents, mostly low-polarity ones, with the dominant one being ethyl acetate (EtOAc), whereas MeOH is the most polar organic solvents used in the extraction/purification schemes. A general scheme of phytoconstituent extraction, purification, and structure identification is presented in Figure 1. A few selected extraction protocols are given below to highlight the similarity and differences in the yield when different solvents were used to extract bisbibenzyls only or alongside terpenes from different bryophytes. For example, air-dried bryophyte samples were extracted with Et_2_O-MeOH [30,31]. This was followed by silica gel or Sephadex LH-20 chromatography using *n*-hexane-EtOAc, C_6_H_6_-EtOAc, or CHCl_3_-MeOH (1:1) as eluents. Then, the eluates were purified with preparative TLC to obtain pure cyclic bisbibenzyls, viz., marchantin A (**1**), B (**2**), and C (**3**), from *Marchantia polymorpha* L., *M. paleacea* var. *diptera* (Nees & Mont.) Inoue, and *M. tosana* Stephani (Figure 2a,b, Figure 3b,d and Figure 4). The yield of the pure compounds ranged from 0.7% to 2.6% of the MeOH-Et_2_O extract [31]. The structures of these isolated compounds were identified using a combination of spectral data, X-ray crystallography and chemical degradation studies [30,31]. In a somewhat similar manner, Huang et al. (2010) extracted a macrocyclic bisbibenzyl from the air-dried powder of *M. tosana* with MeOH, followed by sonication for 1 h and incubation for 2 weeks at room temperature. Then, the extract was filtered, evaporated to dryness, resuspended in water, and partitioned successively with *n*-hexane and EtOAc. The EtOAc fraction exhibited significant anticancer activity against MCF-7 cells. Therefore, it was chromatographed in a silica gel column and purified on a preparative reversed-phase high-performance liquid chromatography (RP-HPLC) with a UV detector. A white amorphous solid was obtained, with a yield of 6.6% of the MeOH extract, which, on subsequent analysis using high-resolution electron ionization mass spectrometry (HR-EI-MS), ^1^H NMR and ^13^C NMR, including 2D NMR, was confirmed to be marchantin A (**1**) [32]. On the other hand, Niu et al., 2006, took the air-dried powder of *M. polymorpha*, and an Et_2_O extract was prepared at room temperature and then concentrated in vacuo. The extract, which showed antifungal activity against *Candida albicans*, was chromatographed on SiO_2_, a stepwise gradient system of petroleum ether (PE)/acetone (Me_2_CO)), followed by rechromatographed on Sephadex LH-20 with mobile phase CHCl_3_/MeOH (1:1) to obtain subfractions, which were further purified with the gradient of PE/Me_2_CO. The purified fractions were confirmed to be marchantin A (**1**), marchantin B (**2**), marchantin E (**4**), neomarchantin A (**5**), 13,13′-*O*-isopropylidenericcardin D (**6**), plagiochin E (**7**) and riccardin H (**8**) through ^1^H NMR and ^13^C NMR [33]. Lu et al., 2006, successfully extracted mixtures of terpenes and macrocyclic bisbibenzyls from air-dried powders of *Conocephalum conicum* (L.) Dumort. (Figure 3a) and *Dumortiera hirsuta* (Sw.) Nees (Figure 3c) following sequential ultrasonic extractions with Et_2_O and MeOH. The Et_2_O extract of *C. conicum* was subjected to adsorption chromatography in silica gel (mobile phase: PE–EtOAc, gradient) to yield seven fractions. These fractions on further chromatographic separation on Sephadex LH-20 and silica gel gave five pure compounds (yield ranging from 0.03% to 0.16% of the Et_2_O extract). The Et_2_O extract of *D. hirsuta* on subsequent chromatography separated into four compounds, including riccardin D (**9**) (yield 0.04% of the Et_2_O extract), whereas the MeOH extract was processed to obtain only two compounds (the yield was 0.04–0.5% of the MeOH extract) [34]. The mixtures of phenolic compounds, cyclic bisbibenzyls, and terpenes were extracted exhaustively by Liu et al., 2011, from the air-dried plant material of *C. japonicum* (Thunb.) Grolle with 95% EtOH under reflux. After resuspending the crude extract in H_2_O and successive partitioning of the extract with Et_2_O and *n*-butanol (BuOH), the Et_2_O extract was subjected to adsorption chromatography on a silica gel column, and eluted with a PE−Me_2_CO gradient system to yield seven fractions. These fractions were separated into eleven pure compounds, with yields ranging from 0.09% to 3.6% of the Et_2_O extract, after further purification using gel permeation, HPLC and other techniques [35].

Owing to the unavailability of bulk samples and the low yield of macrocyclic phenol compounds, it will be useful to have a method that can detect such compounds from a crude sample even if the concentration is very low. In this direction, Xing et al., 2007, developed a rapid screening protocol employing LC-DAD/MS analysis, which could detect even very low concentrations. Xing et al., 2007, extracted the dried powders of *M. polymorpha*, *Asterella angusta* (Stephani) Pandé, K.P. Srivast & Sultan Khan, and *Plagiochasma intermedium* Lindenb.& Gottsche with EtOH, followed by sonication for 30 min and vigorous shaking for complete extraction. The ethanol extracts were partitioned into Et_2_O and water. The Et_2_O fraction was dissolved in acetonitrile (ACN), passed through a microporous membrane (0.45 µm) and subjected to LC-DAD/MS analysis. The method was capable of detecting very low concentrations of bisbibenzyls in a 50 min run. Because of high sensitivity and a relatively short run time, this method could be routinely used for screening macrocyclic bisbibenzyl compounds in bryophytes [36].

Direct molecular analysis from single-cell content was reported by using a nanoelectrospray tip following the analysis of the mass profile of the compound using nanoelectrospray ionization [37]. Isomers can be separated by combining online MS with ion-mobility separation which enables the rapid sampling at sub-attomolar-level sensitivity [38]. Following the same technique, oil body contents were analyzed using a micromanipulator, and direct evidence of two sesquiterpenes and marchantin A (**1**) was detected. The number of oil body cells correlated with the quantified amount of the metabolites, and low-mineral condition induced the increase in oil-body cells in the thallus [39].

Quantities of bibenzyls, bisbibenzyls, and their derivatives in liverworts, along with their spectral characteristics, have been compiled for the first time, as presented in Table 1. Compiled data regarding the quantity of phytochemicals is calculated in 100 g of dry weight for the convenience of presentation. A brief discussion of the tabulated data is presented below to show the quantitative variations of phytochemicals in inter- and intraspecific taxa.

Asakawa et al., 1982, extracted 6.11 g of perrottetin D (**10**) from *Radula perrottetii* Gottsche ex Stephani with diethyl ether following methanol extraction [40], and 0.149 g of perrottetin D was obtained directly in methanol [41], though both the samples were collected from the same geographical region (Tokushima, Japan). Riccardin A (**11**) and B (**12**) production varies from 0.13 g to 4.45 g and 0.11 g to 4.25 g, respectively, in the same species, i.e., *Riccardia multifida* (L.) Gray of the same geographical region and extracted in the same solvent system, as reported by Asakawa et al., 1983, and Nagashima et al., 1996, respectively [42,43]. Perrottetin E (**13**) was extracted at 83.57 mg, 36.82 mg, and 218.4 mg, respectively, from *R*. *perrottetii*, *R. laxiramea* Stephani, and *Pellia epiphylla* (L.) Corda [41,44,45]. The quantity of perrottetinene (**14**) also varies from 18.57 to 562.5 mg in *R. laxiramea* and *R. marginata* Gottsche, Lindenb. & Nees [45,46].

Marchantin A (**1**)was obtained from Hungarian *M. polymorpha* in quantities of 0.14 g and 1.9 g to 5/6 g when extracted *M*. *paleacea* subsp. *diptera* [47,48], and the quantity of marchantin B (**2**) was 0.13 mg to 12.87 mg from *M. polymorpha* [21,33], 13.48 mg from *M. paleacea* subsp. *diptera* [21], and 42.85 mg from *Schistochila glaucescens* (Hook.) A. Evans [49]. Marchantin C (**3**) was obtained in 6.5 mg from *M. polymorpha* [21] and 16.69 mg from *M. paleacea* subsp. *diptera* [21] showing that the quantity of the same phytochemical varies with species or genus or may be influenced by geography and season as well. Bisbibenzyl contents of *P. appendiculatum* Lehm. & Lindenb. vary in soil-grown thallus, and hence the induced callus in the laboratory may be utilized further for the mass production of bisbibenzyl in optimum growth conditions [50].

**Table 1 plants-12-04173-t001:** Total content (in mg/100 mg dry weight) and chemical characteristics of bibenzyl and bisbibenzyl compound reported from bryophytes.

Type of Chemical	Name of the Chemical	Name of the Taxa	Content (mg/100 g dw)	λmax (nm)	Mass Spectra (*m*/*z*)	References
Prebibenzyl	Dumhirone A (**20**)	*Dumortiera hirsuta*	0.4	274 in MeOH	230.0963 [M]^+^	[51]
	Prelunularin(**22**)	*Ricciocarpos natans*	1.2	272	na	[52]
Bibenzyl	Dihydroresveratol (**65**)	*Blasia pusilla*	2.82	na	na	[53]
	Brittonin A (**39**)	*Frullaniabrittoniae* subsp. *truncatifolia*	50	217	362 [M]^+^	[54]
	Brittonin A (**39**)	*Frullania inouei*	2.55	na	na	[55]
	Brittonin B (**40**)	*Frullania brittoniae* subsp. *truncatifolia*	25	276	346 [M]^+^	[54]
	Brittonin B (**40**)	*Frullania inouei*	0.83	na	na	[55]
	3,4′-Dimethoxyl-4-hydroxylbibenzyl (**141**)	*Radula complanata*	11.76–14.11	na	258 [M]^+^,	[56]
	2-(3-Methyl-2-butenyl)-3,5-dihydroxybibenzyl (**32**)	*Radula voluta*	1034.38	284 in MeOH	282 [M]^+^	[57]
	2-Geranyl-3,5-dihydroxybienzyl (**33**)	*Radula voluta*	381.25	284 in MeOH	350 [M]^+^	[57]
	3,5-Dihydroxy-4-(3-hydroxy-3-methylbutyl) bibenzyl (**34**)	*Radula laxiramea*	117.30		300 [M]^+^	[45]
	Radulanolide (**44**)	*Radula complanata*	42.82	216	322 [M]^+^	[40]
	Radulanin H (**48**)	*Radula complanata*	999.14	220	324 [M]^−^	[40]
	Radulanin L (**51**)	*Radula complanata*	147.54	282	296.1446 [M]^+^	[56]
	Lunularin (**21**)	*Blasia pusilla*	3.77	na	na	[53]
	Lunularin (**21**)	*Dumortiera hirsuta*	0.85	na	na	[51]
	Lunularic acid (**23**)	*Blasia pusilla*	0.48	na	na	[53]
Prenyl bibenzyl	Perrottetin A (**52**)	*Radula perrottetii*	11,162.08	211	298 [M-91]^−^	[40]
	Perrottetin D (**10**)	*Radula perrottetii*	6116.20	216	296 [M]^−^	[40]
	Perrottetin D (**10**)	*Radula perrottetii*	149.85	na	296.1425 [M]^+^	[56]
	2-Carbomethoxy-3,5,4″-trihydroxy- 4,6-(3-methyl-2-butenyl)-Bibenzyl (**27**)	*Lethocolea glossophyla*	72	313 in MeOH	424 [M]^+^	[57]
	2,4-di-(3-methyl-2-butenyl)-3,5,4′-Trihydroxy-bibenzyl (**28**)	*Lethocolea glossophyla*	44	313 in MeOH	366 [M]^+^	[57]
	2,2-Dimethyl-5-hydroxy-6-(3-methyl-2-butenyl)-7-[2-(4′-hydroxyphenyl)-ethyl]-chromene (**29**)	*Lethocolea glossophyla*	52	315 in MeOH	364 [M]^+^	[57]
	2,2-Dimethyl-5-hydroxyl-7-[2-(4′-hydroxyphenyl)-ethyl]-8-(3-methyl-2-butenyl)-chromene (**30**)	*Lethocolea glossophyla*	232	320 in MeOH	364 [M]^+^	[57]
Bibenzyl cannabinoid	Perrottetinene (**14**)	*Radula marginata*	562.5	na	na	[46]
	Perrottetinene (**14**)	*Radula laxiramea*	18.57	na	na	[45]
	Perrottetinenic acid (**58**)	*Radula marginata*	87.5	301.4 in EtOH	392.1985, 392 [M]^+^	[46]
Cinnamoyl bibenzyl	Pallidisetin A (**35**)	*Polytrichum pallidisetum*	0.50	318	342.1277 [M]^+^	[58]
	Pallidisetin B (**36**)	*Polytrichum pallidisetum*	0.21	294	342.1236 [M]^+^	[58]
Bibenzyl derivative	Pellepiphyllin (**37**)	*Pellia epiphylla*	6.8	na	258 [M]^+^	[44]
	7-Hydroxy-pellepiphyllin (**38**)	*Pellia epiphylla*	0.24	na	273 [M-H]^−^	[44]
	2,5,4′-Trihydroxy–bibenzyl (**24**)	*Ricciocarpus natans*	0.8	284	na	[52]
	3-Methoxybibenzyl (**26**)	*Radula complanata*	11.76	212	na	[59]
	3,3′,4,4′-Tetramethoxy-bibenzyl (**41**)	*Frullania inouei*	0.34	na	na	[55]
	Chrysotobibenzyl (**42**)	*Frullania inouei*	9.02	na	na	[55]
Bisbibenzyl and its derivative	Riccardin A (**11**)	*Riccardia multifida*	130.64	213	438.1831	[42]
	Riccardin A (**11**)	*Riccardia multifida*	4458.33	283	na	[43]
	Riccardin B (**12**)	*Riccardia multifida*	115.64	233	424.1674	[42]
	Riccardin B (**12**)	*Riccardia multifida*	4250	280	424.1700	[43]
	Riccardin C (**15**)	*Plagiochasma intermedium*	0.22 ^a^	na	na	[60]
	Riccardin C (**15**)	*Marchantia palmata*	9	na	na	[61]
	Riccardin C (**15**)	*Blasia pusilla*	53.35 3.74	na	na	[53]
	Riccardin C (**15**)	*Dumortiera hirsuta*	0.23	na	425.30 [M^−^]	[51]
	Riccardin D (**9**)	*Monoclea forsteri*	141	na	424.1674	[62]
	Isoriccardin C (**83**)	*Plagiochasma intermedium*	0.05 ^a^	na	na	[60]
	Isoriccardin C (**83**)	*Marchantia palmata*	12.5	na	na	[61]
	Isoriccardin C (**83**)	*Marchantia paleacea*	4.31	na	na	[63]
	Isoriccardin D (**87**)	*Marchantia polymorpha*	0.45	na	424.16 [M]^+^	[64]
	Riccardin F (**79**)	*Plagiochasma intermedium*	0.07 ^a^	na	na	[60]
	Riccardin F (**79**)	*Blasia pusilla*	14.64	215	438 [M]^+^	[53]
	Riccardin H (**8**)	*Marchantia polymorpha*	0.06	na	456 [M]^+^	[33]
	Riccardin I (**82**)	*Asterella angusta*	0.32	na	454.1440 [M]^+^	[65]
	Paleatin A (**95**)	*Marchantia paleacea* subsp. *diptera*	5.75	na	na	[21]
	Paleatin B (**96**)	*Marchantia paleacea* subsp. *diptera*	16.51	na	na	[21]
	Polymorphatin A (**136**)	*Marchantia polymorpha*	0.40	na	424 [M]^+^	[64]
	Marchantin A (**1**)	*Marchantia polymorpha*	0.13	na	439 [M]^+^	[33]
	Marchantin A (**1**)	*Marchantia polymorpha*	377.5	na	na	[21]
	Marchantin A (**1**)	*Marchantia paleacea* subsp. *diptera*	1204.54	na	na	[21]
	Marchantin A (**1**)	*Marchantia paleacea* subsp. *diptera*	5000–6000	na	na	[47]
	Marchantin A (**1**)	*Marchantia paleacea* subsp. *diptera*	1191.90	na	na	[48]
	Marchantin A (**1**)	*Marchantia tosana*	na	na	440.1632	[32]
	Marchantin B (**2**)	*Marchantia polymorpha*	0.13	na	455 [M]^+^	[33]
	Marchantin B (**2**)	*Marchantia polymorpha*	12.87	na	na	[21]
	Marchantin B (**2**)	*Marchantia paleacea* subsp. *diptera*	13.48	na	na	[21]
	Marchantin C (**3**)	*Marchantia paleacea* subsp. *diptera*	16.79	na	na	[21]
	Marchantin C (**3**)	*Marchantia polymorpha*	6.5	na	na	[21]
	Marchantin C (**3**)	*Marchantia polymorpha*	40	208.5	na	[61]
	Marchantin C (**3**)	*Marchantia palmata*	10	na	na	[61]
	Marchantin C (**3**)	*Dumortiera hirsuta*	0.3	na	na	[60]
	Marchantin C (**3**)	*Schistochila glaucescens*	61.22	na	na	[49]
	Marchantin C (**3**)	*Reboulia hemisphaerica*	120	na	424.1680 [M]^+^	[66]
	Marchantin C (**3**)	*Marchantia paleacea*	38.79	na	na	[63]
	Isomarchantin C (**67**)	*Marchantia polymorpha*	8.63	211.5	424.1707	[61]
	Isomarchantin C (**67**)	*Marchantia palmata*	6	na	na	[21]
	Marchantin D (**147**)	*Marchantia paleacea* subsp. *diptera*	5.30	na	na	[21]
	Marchantin E (**4**)	*Marchantia paleacea* subsp. *diptera*	126.36	na	na	[21]
	Marchantin E (**4**)	*Marchantia polymorpha*	0.11	na	469	[33]
	Marchantin E (**4**)	*Marchantia paleacea* subsp. *diptera*	125.03	na	na	[48]
	Marchantin H (**144**)	*Plagiochasma intermedium*	0.04 ^a^	na	na	[60]
	Marchantin M (**72**)	*Reboulia hemisphaerica*	100	283	454.1774 [M]^+^	[66]
	Marchantin N (**74**)	*Reboulia hemisphaerica*	120	276.5	471 [M]^+^	[66]
	Marchantin O (**75**)	*Reboulia hemisphaerica*	300	274	438.1817 [M]^+^	[66]
	Marchantin J (**68**)	*Marchantia polymorpha*	30.30	na	424 [M]^+^	[64]
	Neomarchantin A (**5**)	*Plagiochasma intermedium*	0.05	na	na	[60]
	Neomarchantin A (**5**)	*Marchantia polymorpha*	0.04	na	423 [M]^+^	[33]
	Neomarchantin A (**5**)	*Schistochila glaucescens*	51.02	272 in EtOH	424 [M]^+^	[49]
	Neomarchantin B (**77**)	*Schistochila glaucescens*	42.85	272 in EtOH	440 [M^+^	[49]
	Pakyonol A (**145**)	*Plagiochasma intermedium*	0.40 ^a^	na	na	[60]
	Angustatin A (**146**)	*Asterella angusta*	0.28	na	436.1330 [M]^+^	[65]
	Isoplagiochin A (**62**)	*Plagiochila* sp.	2.85	288–250 Sh	423 [M+H]^+^	[67]
	Isoplagiochin A (**102a**)	*Plagiochila fruticosa*	33.48 ^a^	na	na	[68]
	Isoplagiochin A (**102a**)	*Heteroscyphus planus*	0.29	292	422.1542 [M]^+^	[69]
	Isoplagiochin B (**103a**)	*Plagiochila fruticosa*	17.79 ^a^	na	na	[68]
	Isoplagiochin C (**104**)	*Herbertus sakerraii*	4.36	na	na	[70]
	Isoplagiochin C (**104**)	*Plagiochila fruticosa*	8.34 ^a^	287	422.1485 [M]^+^	[68]
	Isoplagiochin D (**105**)	*Herbertus sakerraii*	35.43	na	na	[70]
	Isoplagiochin D (**105**)	*Plagiochila fruticosa*	5.32 ^a^	na	424.1654 [M]^+^	[68]
	Isoplagiochin F (**107**)	*Plagiochila* sp.	2.85	283–251 Sh	441 [M+H]^+^	[67]
	Dihydroisoplagiochin (**108**)	*Plagiochila* sp.	5	288–282 Sh	425 [M+H]^+^	[67]
	Monochlorinated isoplagiochin D (**109**)	*Plagiochila* sp.	0.71	288–282	460	[67]
	2,12-Dichloroisoplagiochin D (**110**)	*Herbertus sakuraii*	6.48	288	492.0904	[70]
	12,7′-Dichloroisoplagiochin D (**111**)	*Herbertus sakuraii*	3.04	288	492.0923	[70]
	12, 10′-Dichloroisoplagiochin C (**112**)	*Herbertus sakuraii*	4.36	294	490.0714	[70]
	12-Chloroisoplagiochin D (**113**)	*Mastigophora diclados*	0.94	na	na	[70]
	2,12-Dichloroisoplagiochin D (**110**)	*Mastigophora diclados*	0.68	na	na	[70]
	Plagiochin E (**7**)	*Marchantia polymorpha*	13.40	na	424.1685	[33]
	Planusin A (**106**)	*Heteroscyphus planus*	3.23	297	422.1485 [M]^+^	[69]
	Pusillatin A (**90**)	*Blasia pusilla*	0.65	212 in EtOH	869 [M+Na]^+^, 846 [M]^+^	[53]
	Pusillatin B (**91**)	*Blasia pusilla*	5.18	227 in EtOH	869 [M+Na]^+^, 846 [M]^+^	[53]
	Pusillatin C (**92**)	*Blasia pusilla*	19.69, 3.05	215 in EtOH	869 [M+Na]^+^, 846 [M]^+^	[53]
	Perrottetin E (**13**)	*Radula perrottetii*	9.62	na	na	[71]
	Perrottetin E (**13**)	*Radula perrottetii*	83.57	na	na	[56]
	Perrottetin E (**13**)	*Plagiochila* sp.	1.42	na	na	[67]
	Perrottetin E (**13**)	*Pellia epiphylla*	218.4	na	na	[44]
	Perrottetin E (**13**)	*Marchantia polymorpha*	2.02	na	424.16 [M]^+^	[64]
	Perrottetin E (**13**)	*Radula laxiramea*	36.82	na	na	[45]
	Perrottetin F (**55**)	*Radula perrottetii*	63.55	na	na	[71]
	Perrottetin F (**55**)	*Radula perrottetii*	86.45	na	na	[56]
	Isoperrottetin A (**56**)	*Radula perrottetii*	11.96	213	426 [M]^+^	[71]
	Perrottetin G (**57**)	*Radula perrottetii*	12.79	na	na	[71]
	Perrottetin G (**57**)	*Radula perrottetii*	4.61	na	na	[56]
	10′-Hydroxyperrottetin E (**59**)	*Pellia epiphylla*	139.44	na	442 [M]^+^	[44]
	10′-Hydroxyperrottetin E-11-methylether (**60**)	*Pellia epiphylla*	11.38	na	456 [M]^+^	[44]
	14′-Hydroxyperrottetin E (**61**)	*Pellia epiphylla*	17.47	na	442 [M]^+^	[44]
	10,10′-Dihydroxy-perrottetin E (**62**)	*Pellia epiphylla*	81.73	na	458 [M]^+^	[44]
	13′,13″-Bis(I0′-hydroxyperrottetin E (**63**)	*Pellia epiphylla*	2.04	na	882 [M]^+^	[44]
	Perrottetin E-11-methyl ether (**64**)	*Pellia epiphylla*	27.44	na	440 [M]^+^	[44]
	Marchantin G methyl ether (**78**)	*Marchantia palmata*	1.5	na	na	[61]
	6′,6″-Bis-riccardin C (**25**)	*Ricciocarpus natans*	4	847	na	[52]
	Isoriccardinoquinone A (**84**)	*Marchantia paleacea*	10.34	na	468 [M]^+^	[63]
	Isoriccardinoquinone B (**85**)	*Marchantia paleacea*	7.75	na	452 [M]^+^	[63]
	2-Hydro-3,7-dimethoxyphenanthrene (**86**)	*Marchantia paleacea*	19.82	na	na	[63]
	13, 13′-*O*-Isopropylidenericcardin D (**6**)	*Marchantia polymorpha*	0.06	na	464	[33]
	11-*O*-Demethyl-marchantin I (**143**)	*Asterella angusta*	0.4	210 in MeOH	423 [M-H]^−^, 424.1712 [M]	[72]
	Dihydroptychantol A (**135a**)	*Asterella angusta*	0.32	210 in MeOH	423 [M-H]^−^, 424.1720 [M]	[72]
	Ptychantol A (**132**)	*Ptychanthus strictus*	39.67	342	HRMS *m*/*z* (422.15)	[73]
	Ptychantol B (**133**)	*Ptychanthus strictus*	21.08	341	HRMS *m*/*z* (438.14)	[73]
	Ptychantol C (**134**)	*Ptychanthus strictus*	1.95	342	HRMS *m*/*z* (438.14)	[73]
Dibenzofuran bisbibenzyl	Asterellin A (**88**)	*Asterella angusta*	0.4	212 in MeOH	421 [M-H]^−^, 422.1514 [M]^+^	[72]
	Asterellin B (**89**)	*Asterella angusta*	0.6	na	435 [M-H]^−^, 436.1655 [M]^+^	[72]
Chlorinated bisbibenzyl	Bazzanin A (**114**)	*Bazzania trilobata*	0.38	250	456 [M]^+^	[74]
	Bazzanin B (**115**)	*Bazzania trilobata*	2.50	255	490.0728 [M]^+^	[74]
	Bazzanin B (**115**)	*Bazzania trilobata*	0.44	na	na	[75]
	Bazzanin C (**116**)	*Bazzania trilobata*	1.13	252	524.0345 [M]^+^	[74]
	Bazzanin D (**117**)	*Bazzania trilobata*	0.93	250	524.0344 [M]^+^	[74]
	Bazzaninin E (**118**)	*Bazzania trilobata*	1.07	252	557.9968 [M]^+^	[74]
	Bazzanin F (**119**)	*Bazzania trilobata*	0.21	255	558 [M]^+^	[74]
	Bazzanin G (**120**)	*Bazzania trilobata*	0.65	255	592 [M]^+^	[74]
	Bazzanin H (**121**)	*Bazzania trilobata*	0.21	252	592 [M]^+^	[74]
	Bazzanin I (**122**)	*Bazzania trilobata*	0.26	252	626 [M]^+^	[74]
	Bazzanin J (**123**)	*Bazzania trilobata*	0.51	252	492.0898 [M]^+^	[74]
	Bazzanin K (**124**)	*Bazzania trilobata*	0.67	246	502 [M]^+^	[74]
	Bazzanin S (**131**)	*Bazzania trilobata*	2.33	na	na	[75]
Bisprenylated bisbibenzyl	Glossophylin (**31**)	*Lethocolea glossophyla*	20	320	729 [M+H]^+^	[57]
Derivative of bisbibenzyl with sesquiterpene moiety	Glaucescenolide (**137**)	*Schistochila glaucescens*	44.89	217 in EtOH	250.1576 [M]^+^	[49]
	GBBA ^b^ (**138**)	*Schistochila glaucescens*	12.24	273 in EtOH	679.3044 [M]^+^	[49]
	GBBB ^b^ (**139**)	*Schistochila glaucescens*	10.20	271 in EtOH	679.3022 [M]^+^	[49]

Foot notes: ^a^ fresh weight ^b^ glaucescenolide, na—data not available, EtOH—ethanol, MeOH—methanol.

## 3. The Occurrence and Diversity of Bibenzyls and Bisbibenzyls in Marchantiophyta

The natural product chemistry of bryophytes has been popularized by Asakawa and his collaborators in the last four decades through the discovery of significant molecules of bibenzyl and bisbibenzyl classes. These are small phenolic molecules with two to four aromatic rings, where hydroxyl or alkoxy substituent groups are present, and these compounds are further elaborated via dimerization or additional ring formation through halogenations or oxidation [18,19,21]. The first discovered bisbibenzyl compounds were marchantin A (**1**), by Asakawa, 1982, and riccardin A (**11**), by Asakawa et al., 1983 [42]. A series of bisbibenzyls have been reported since then, and about 70 bisbibenzyl compounds have been characterized [15,19,20,21]; macrocyclic bisbibenzyls are exclusive chemical compounds of liverworts but are exceptionally reported from moss, i.e., marchantin C, M, N, O from *Reboulia hemispherica* (Figure 5) [66], and riccardin C (**15**) was recently identified in the acetone extract of *Primula macrocalyx* Bunge when fractionated through column chromatography. This is the first bisbibenzyl from flowering tracheophytes [76], and the acyclic bisbibenzyl, perrottetin H (**16**), was also reported earlier from a fern *Hymenophyllum barbatum* (Bosch) Baker [77]. Bisbibenzyls have different structural moieties, and it is absolutely necessary that the conformations of the molecule are involved in the reactions—this structure–activity relation is discussed below. The structure of bibenzyls, prebibenzyls, and stilbenoid compounds is covered in Figure 6, while the structure of some important macrocyclic and acyclic bisbibenzyls are presented in the structure–activity section in Figure 7, Figure 8, Figure 9, Figure 10, Figure 11, Figure 12, Figure 13 and Figure 14.

### 3.1. Diversity of Bibenzyls and Prebibenzyls

Bibenzyls are compounds with the dihydro-stilbenoidF type skeleton, and they are abundantly distributed in liverworts and serve as chemotaxonomic markers. Macrocylic bisbibenzyls and acyclic bisbibenzyls are derived from bibenzyls (see the next section). The following two consecutive sections describe the critical analysis of different bibenzyl and bisbibenzyls with their source taxa and structure elucidation through derivative formation following the spectral characteristics and mass profile most extensively used by researchers, as shown in Table 1.

Prebibenzyls, bibenzyls, prenylated bibenzyls, cavicularin and cannabinoid bibenzyls have varying carbon skeletons and are distributed in liverworts, as well as some angiosperm families. *Plagiochila buchtiniana* Stephani, *P. diversifolia* Lindenb. & Gottsche and *P. longispina* Lindenb. & Gottsche produce three prebibenzyls, which were also later reported from the New Zealand liverwort *Balantiopsis rosea* Berggr. The structures of these three prebibenzyls (longispinone A (**17**), longispinone B (**18**), and longispinol (**19**), Figure 7)) were confirmed using different spectroscopic techniques as (5*S*)-*p*-(methoxyphenylethyl) cyclohexan-1-one, (5*R*)-*p*-(methoxyphenylethyl) cyclohexan-1-ol and (5*R*)-*p*-(methoxyphenylethyl) cyclohex-2-en-1-one. These three compounds are probable precursors of the dicarboxylic bibenzyls or some macrocyclic bisbibenzyls [78]. An unusual phenylethyl cyclohexadienone was identified from *D. hirsuta* as dumhirone A (**20**) (Figure 7), the only reported natural product containing a 4-substituted-cyclohexa-2,5-dienone skeleton, and it can be synthesized from lunularin (**21**). The occurrence of dumhirone in *D. hirsuta* belonging to the family Marchantiaceae indicates the possible relationship of the biosynthetic pathways with genera like *P. longispina* (Plagiochilaceae) and *B. rosea* (Balantiopsidaceae) accumulating hydrogenated bibenzyl derivatives, though experimental evidence is required [51].

Lunularin (**21**, Figure 7) is a prebibenzyl obtained from *M. polymorpha*, *Ricciocarposnatans* and *C. conicum* [79]. Prebibenzyl prelunularin (**22**) (Figure 4) gives rise to lunularin, which, along with lunularic acid (**23**) acts as a precursor for bisbibenzyl synthesis, and it has been identified in many liverworts, including *M. polymorpha*, *M. tosana*, *B. pusilla*, *Frullania convoluta* Lindenb. & Hampe, and *Monoclea forsteri* Hook. [79] (see Figure 7). Bibenzyl 2,5,4′-trihydroxybibenzyl (**24**), a dimeric bisbibenzyl (6′,6″-bis-riccardin C, **25**) and a phenylethylcyclohexenone (prelunularin, **22**) were obtained from an axenic culture of *Ricciocarpos natans* [52]. The contents of *Radula complanata* (L.) Dumort. oil bodies were analyzed using GC-MS, and 3-methoxybibenzyl (**26**) was found as the main constituent [59].

Prenylated bibenzyl derivatives were obtained from the liverwort *Lethocolea glossophylla* (Spruce) Grolle, and their structures were elucidated through spectral analysis as 2-carbomethoxy-3,5,4″-trihydroxy-4,6-(3-methyl-2-butenyl)-bibenzyl (**27**); 2,4-di-(3-methyl-2-butenyl)-3,5,4′-trihydroxybibenzyl (**28**); 2,2-dimethyl-5-hydroxy-6-(3-methyl-2-butenyl)-7-[2-(4′-hydroxyphenyl)-ethyl]-chromene (**29**); 2,2-dimethyl-5-hydroxy-7-[2-(4′-hydroxyphenyl)-ethyl]-8-(3-methyl-2-butenyl)-chromene (**30**); and, with these, one bisprenylated bisbibenzyl named glossophylin (**31**). Two bibenzyls, 2-(3-methyl-2-butenyl)-3,5-dihydroxy-bibenzyl (**32**) and 2-geranyl-3,5-dihydroxy-bibenzyl (**33**), were identified from *Radula voluta* Taylor [57]. A new prenyl bibenzyl named 3, 5-dihydroxy-4-(3-hydroxy-3-methylbutyl) bibenzyl (**34**) was reported from *Radula laxiramea* [45]. Two cinnamoyls, bibenzyl pallidisetin A (**35**) and pallidisetin B (**36**), were reported from a moss *Polytrichum pallidisetum* Funck [58]. Two bibenzyl derivatives pelliepiphyllin (**37**) and 7-hydroxypelliepiphyllin (**38**) were obtained from *P. epiphylla* [44]. Two bibenzyls, brittonin A (**39**) and brittonin B (**40**), and two bibenzyl derivatives, 3, 3′-4,4′-tetramethoxybibenzyl (**41**) and chrysotobibenzyl (**42**), were obtained from *Frullania inouei* S. Hatt. [55], though the bibenzyls were earlier reported from *Frullania brittoniae* subsp. *truncatifolia (Stephani)* R. M. Schust. & S.Hatt. [54].

*Cavicularia densa* Stephani (family Blasiaceae) produces a novel optically active cyclic compound (+) cavicularin (**43**) (Figure 7), which is one of the unusual natural products with a bibenzyl and dihydrophenanthrene unit conjugated via a biaryl bond and an ether linkage. This compound lacks a stereogenic carbon center but, possessing conformational chirality, is one of the major attractions of synthetic chemists. This compound might be formed with intramolecular phenolic oxidative coupling between C-3′ and C-10′ from riccardin C (**15**) [18].

Radulanolide (**44**), a bibenzyl with a chemical formula of C_20_H_18_O_4_, was identified from *Radula complanata* [40]. Radulanin A (**45**) and radulanin C (**46**) are seven-membered heterocyclic rings and are richly present in *R. buccinifera* (Hook.f. & Taylor) Gottsche, Lindenb. & Nees, *R. complanata*, *R. japonica* Gottsche ex Stephani, *R. tokiensis* Stephanii and *R. variabilis* S. Hatt., among which the last four species also produce radulanin E (**46**) [40]. Radulanin H (**48**) is extracted from *R. complanata* and *R. perrottetii* [40]. Two bibenzyls with a dihydrooxepin skeleton, radulanin A-5-one (**49**) and β-hydroxyradulanin 5-one (**50**), were reported in *Marsupidium epiphytum* Colenso [19]. The morphological separation of different sections of the genus *Radula* is supported by chemical evidence; i.e., prenylated bibenzyls or bibenzyl derivatives with dihydro-oxepin skeleton have only been noted in *Radula* species. So, these bibenzyls are significant chemosystematic markers of the family Radulaceae. Angiosperm families Compositae and Leguminosae also possess such bibenzyls [40,79]. Radulanin L (**51**) was reported from *Radula complanata* [56].

Perrottetins A-D (**52**–**54**, **10**), perrottetin E (**13**) and perrotetin F (**13**) were first reported from *R. perrottetii* [40]. Perrottetin H (**16**) is an acyclic bisbibenzyl obtained from Marchantiophyta and used as a chemosystematic marker [79]. Isoperrottetin A (**56**), a novel bisbibenzyl obtained from *R. perrottetii* [71]. *R. perrottetii* and *Radula marginata* produce (-) perrottetinene (**14**) (Figure 7), a cannabinoid derivative with the *cis* moiety structurally analogous to tetrahydrocannabinol from *Cannabis sativa* L. PerrottetinG (57) and perrottetinenic acid (**58**) were also obtained from *Radula perrottettii* and *Radula marginata* [46,56,71]. Bibenzyl tetrahydrocannabinoids, which were previously reported from higher plants, have *trans* geometric isomerism at the cyclohexane ring. Apart from the structural analogy, some shared features have been observed in *C. sativa* L. and *Radula marginata*; that is, the cannabinoids in *C. sativa* are stored in specialized trichome, whereas in *R. marginata*, oil bodies are the depository sites of these secondary metabolites [80]. 10′-Hydroxyperrottetin E (**59**), 10′-hydroxyperrottetin E-11-methylether (**60**), 14′-hydroxyperrottetin E (**61**), 10, 10′-dihydroxyperrottetin E (**62**), 13′, 13″-bis(O′-hydroxyperrottetin E) (**63**) and perrottetin E-11-methyl ether (**64**) were isolated from *P. epiphylla* [44].

Dihydroresveratrol (**65**) (resveratrol, a well-known stilbenoid compound of red wine with antineoplastic properties; Figure 7), previously isolated from *Dioscorea dumentorum*, was also reported from *Blasia pusilla* of Metzgeriales [19].

### 3.2. Diversity of Macrocyclic and Acyclic Bisbibenzyls

Bisbibenzyls are the dimeric form of bibenzyls present in cyclic or acyclic forms in different orders of Marchantiophyta, and their derivatives are also naturally present in some taxa of liverworts, as seen in Table 1. Naturally produced from two molecules of lunularin (21), each has a core structure consisting of four aromatic rings and two ethano bridges. Marchantin type bisbibenzyls are more diverse and predominantly reported from different species of *Marchantia* and also from some other genera, and they act as chemotaxonomic markers [15,19].

#### 3.2.1. Marchantins, Isomarchantins and Their Derivatives

Marchantins are a class of bisbibenzyl compounds with two diaryl ether subunits. The biosynthesis of the diaryl ether natural products is evolving continuously. Their structural analogs are given much importance due to their pharmacological implications [18,21,32,81,82].

The oxygen atom in diaryl ether is a favorable flexible linker that can allow molecules to form multi-dimensional conformation, and due to the binding affinity to a potential target, its importance in molecular drug design is increasing [83]. Marchantins were the first reported bisbibenzyls [40] among all the macrocyclic bisbibenzyls, and they represent the largest class of bisbibenzyls reported from different species of the genus *Marchantia* (*M. polymorpha*, *M. palmata*, *M. diptera* subsp. *paleacea* and others) and also from the genera *P. intermedium*, *Reboulia hemisphaerica*, etc. (Figure 5) [30,61,66]. The structures of the marchantin series compounds were determined through ^1^H and ^13^C NMR, and COSY. The structure of marchantin A (**1**) was determined through chemical degradation and X-ray crystallographic analysis of its trimethyl ether (**66**). Protonated aromatic carbons were identified using the ^13^C-^1^H correlation spectrum. The remaining aromatic and benzylic carbons are determined using long-range selective proton decoupling (LSPD) experiments. As the other compounds of the marchantin series are non-crystaline and available in very small quantities, the total assignment of ^1^H-^13^C would facilitate the detection of the other members of this class [21,31,60,61]. The ^1^H NMR spectrum showing the characteristic signal appearing at δ 5.13 (J = 2 Hz) is due to H-3′ strongly shielded by ring A [31], whereas the proton signal for marchantin C (**3**) and isomarchantin C (**67**) appears at δ 5.52 and δ 6.08, respectively [61]. Marchantin A (**1**) and its trimethyl ether (**66**) are difficult to crystallize as they are tightly protected from water/solvents. To eliminate the crystallization difficulty, a small column packed with silica gel and dried magnesium sulfate (1:1) was prepared, and each sample passed through this column gives a beautiful single crystal [21]. The total synthesis of marchnatin A (**1**) was accomplished after its discovery from the natural product [84]. The Indian and French races of *M. polymorpha* produce marchantin E (**4**) as major components, whereas Japanese *M. polymorpha* produces marchantin A (**1**) as a major component [61,85]. Marchantin B (**2**) and C (**3**) were also discovered [30] with two ether links from *M. polymorpha*, *M. paleacea* subsp. *diptera* and *M. tosana*. A few years later, Asakawa and coworkers reported Marchnatin J-L (**68**–**70**) from the German race of *M. polymorpha*. Marchantin J trimethyl ether (**71**) possessed an ethoxyl group resembling marchantin E. ^1^H and ^13^C NMR spectroscopy, including NOE difference spectra, confirmed the structure of the derived compound, and this is the first record of the ethoxylated compound from liverworts [86]. The ^1^H and ^13^C NMR signals of marchantin M (**72**) were very close to those of marchantinquinone (**73**) except ^13^C absorption on the B-ring and two protons (δ 6.62, d and 6.70, *d*, *J =* 8.6 Hz) on ring B, and the two hydroxyl and methoxy substituents must be located on either C-10 or C-13, as well as a strong molecular ion *m*/*z* at 454. Final assignments were based on the hydroxyl group at C-10 based on the results of deuterium-induced ^13^C chemical shifts, and the structure was defined as marchantin M (**72**). Another similar compound giving the singlet proton peak at δ 5.73 on the B-ring suggested the methoxy group at either C-11 or C-12. The functional group at C-10 and C-13 was either a ketone (benzoquinone) or a hydroxyl (hydroquinone). ^13^C shifts of the compound were assigned to marchantin N (**74**), as the placement of the methoxy group at C-11 was mainly based on the shifts of C-10 and C-13 [66]. Marchantin O (**75**) was purified from Taiwanese liverwort *R. hemisphaerica* and characterized using NMR spectra and earlier isolated and identified from Japanese *R. hemisphaerica* [66]. Marchantin P (**76**) was characterized based on *M. chenopoda* L. of Venezuela [87]. Isomarchantins have a similar structure to marchantins, though the connectivity between arenes B and D is opposite to that found in the marchantins. Isomarchantin C (**67**) was isolated from Indian *M. polymorpha* and *M. palmata* Reinw., Nees & Blume, followed by structural characterization by the ^1^H and ^13^C NMR spectra [61]. Neomarchantin A (**5**) and B (**77**), with diaryl ether subunits, were first isolated from a leafy liverwort *S. glaucescens* and a thalloid liverwort *Preissia quadrata* (Scop.) Nees [49]. In these compounds, two para- and meta-substituted arenes form the macrocycle and impart some unusual bond angles to the natural products in he solid state. The structure elucidation was carried out using COSY, HMQC, HMBC and NOE data [49].

Air-dried *M. palmata* was extracted with diethyl ether, and the ether-soluble portion was treated with alkali and neutralized with diluted HCl and further separated with column chromatography. Bibenzyl mixtures of the same R_f_ value were obtained from the fraction further methylated to form marchantin G methyl ether (**78**), which was confirmed through ^1^H NMR [61].

#### 3.2.2. Bisbibenzyls of Riccardin Family and Derivatives

The Riccardin family of compounds has 18 membered rings, and its members are characterized by one diaryl ether, one biaryl linkage, and a para-disustituted A ring. Benzene units are connected via ether oxygen between rings A and B and a biphenyl bond between the benzene rings C and D, which has been reported since the 1980s. *Riccardia multifida* subsp. *decrescens* produces riccardins A and B (**11**–**12**) [30]. Riccardin C (**15**) was isolated from *B. pusilla*, *D. hirsuta*, *Jungermannia infusca* (Mitt.) Stephani and *M. paleacea* subsp. *diptera*, *Mastigophora diclados*, *Monoclea forsteri*, *Plagiochasma pterospermum* C. Massal, *P. rupestre* (G.Frost) Stephani, *R. hemisphaerica* and *Ricciocarpus natans*, riccardin F (**79**) from *M. tosana*. Riccardin C was the first macrocyclic bisbibenzyls reported from the liverwort [88] and a higher plant [76]. The structure of riccardin A (**11**) was confirmed based on the formation of its diacetate and trimethyl ether derivatives [30]. The ^1^H NMR spectrum of riccardin A (**11**), a strongly shielded one-proton doublet at δ 5.33 (^1^H). has been assigned to an Inner proton H-3′ on benzene ring C, which lies over the plane of benzene ring B [30]. The structure of riccardin B (**12**) was determined based on a comparison of the ^1^H and ^13^C NMR spectra with those of riccardin A (**11**) and its derivatives. The methylation of riccardin C (**15**) gave trimethyl ether, whose spectral data were identical to the dimethyl ether of riccardin A [88]. The structure elucidation of riccardin D (**9**), riccardin E (**80**), riccardin F (**79**) and riccardin G (**81**) was performed through extensive decoupling and NOE spectrometry on the naturally occurring compounds and their premethylated products [30,87]. Riccardins A and B may be biosynthesized from lunularic acid (**23**) or lunularin (**21**), which are widespread in thalloid liverworts [30]. Riccardin family members were synthesized through a convergent scheme [89]. Another riccardin family member, riccardin H (**8**), and 13,13′-*O*-isopropylidenericcardin D (**6**) were reported from Chinese *M. polymorpha* [33]. Another novel variant, benzyloxidized bis (bibenzyl), belonging to riccardin and named riccardin I (**82**), was isolated from *A. angusta* [65]. Macrocyclic bisbibenzyl mono ether was reported for the first time from *M. palaeacea* [63]. Phenolic protons are ascribed based on the broad singlet at δ 5.37 and 5.84 and the quinone carbonyl carbons at δ 187.4 and 187.5 from ^13^C NMR spectra. The presence of the bisbibenzyl moiety was indicated by the appearance of four characteristic carbon peaks at δ 31.0, 33.9, 33.9 and 38.1 in ^13^C NMR spectra, and the ether oxygen linked to two benzene rings is further confirmed by DEPT and HMBC correlation.

Isoriccardin C (**83**) was reported from Indian *M. polymorpha* and *M. palmata* [61]. Hong Kong liverwort *M. paleacea* thallus in 95% ethanol gives rise to two novel isoriccardinquinones, A (**84**) and B (**85**), along with marchantin C (**3**), isoriccardin C (**83**) and a phenanthrene derivative, 2-hydroxy-3,7-dimethoxyphenanthrene (**86**), whose structures were confirmed using 2D NMR [63]. Isoriccardin D (**87**), along with other bisbibenzyl derivatives, was identified from Chinese *M. polymorpha* in ethanol extract [64].

Asterelins A and B (**88**–**89**), the first dibenzofuran bisbibenzyls obtained from *A. angusta*, and the elucidation of their structure were established using 1D and 2D NMR, MS and XRD. The coupling patterns of proton resonances in the range from δ_H_ 5.33 to 7.47 established the presence of four independent aromatic rings. The four signals at δ_H_ 6.61, 7.47, 5.98, and 6.29, could be assigned to 1,4-disubstituted benzene ring (ring A) from their coupling patterns using the ^1^H-^1^H COSY, while the two resonances at δ_H_ 6.83 and 6.87 suggest the presence of a 1,2,3,5-tetrasubstituted benzene ring (ring B). The signals at δ_H_ 5.33, 6.84 and 6.84, together with resonances at δ_H_ 6.28, 7.14, and 7.26, indicated the presence of 1,2,4-trisubstituted benzene (ring C and D, respectively). Both the compounds contained a novel dibenzofuran linkage, and the stereochemical structure of asterelin B (**89**) suggested the presence of a magnetically anisotropic effect in ring A; the compounds are present as a natural product, as confirmed via the HPLC-UV analysis of cold diethyl ether extract, not as an artifact [72].

#### 3.2.3. Pusilatins-Derivatives of Riccardin

*B. pusilla* L., another genus of the family Blasiaceae, produces riccardin C (**15**) dimer, i.e., pusilatins A, B, C and D (**90**–**93**), which are also produced through oxidative coupling reactions [53]. Another novel structure was elucidated in the consecutive year, and the monomethyl ether of pusilatin B (**91**) was named pusilatin E (**94**) isolated from *Riccardia multifida* subsp. *decrescens* [53]. In pusilatin A (**90**), four benzyl methylene signals (δ 35.7, 37.9, 38.6 and 38.8) were detected in the ^13^C NMR. The ^1^H NMR spectrum indicated the presence of 12 protons on benzene rings at δ 5.37–7.14 ppm and four benzylic methylenes at δ 2.61–2.92 ppm (8H). Spectral data show that pusilatin A is a dimer of riccardin C with a C12-C-12″ phenyl linkage. Pusilatin A (**90**) shows a similar coupling pattern with riccardin C, and the structure was established with a 2D NMR spectra, including ^1^H-^1^H COSY, HMQC and HMBC. The stereostructure of pusilatin A was established using the X-ray crystallographic analysis of its hexaacetate derivatives [53]. Also, the spectral data of pusilatin B were almost the same as those of pusilatin A. Pusilatin C (**92**) is an asymmetrical dimer of riccardin C (**15**). Pusilatin D (**93**) is regarded as a dimeric riccardin C linked by an ether C-12-*O*-C-1‴ bond. The ^13^C NMR spectrum exhibited 56 signals, including 8 benzyl methylene signals (δ 37.8, 37.89, 37.92 (two carbons), 38.3, 38.5, and 38.7 (two carbons)). Methylated and acetylated derivatives further confirm the location of functional groups and linkage patterns [53].

#### 3.2.4. Paleatins

Two acyclic bisbibenzyls were obtained from *M. paleacea* subsp. *diptera*, i.e., paleatin A and paleatin B (**95**–**96**), where rings A and C, and B and D, are connected through oxygen bonds. Paleatin B (**96**) differs from A due toan additional methyl group [21].

#### 3.2.5. Plagiochin Type Bisbibenzyl

Plagiochins A, B, C and D (**97**–**100**), first isolated from *Plagiochila sciophila* Nees (=*Plagiochila acanthophylla* subsp. *japonica*), possess two *ortho*-biphenyl linkages between two benzyl groups. ^1^H, ^13^ C NMR and NOE studies, as well as XRD analysis on its tetramethyl ether, showed that ring A is perpendicular to ring C and parallel with ring D. The proton at −3′ is strongly shielded by both rings A and D, causing a high field shift to δ 4.84 in the tetramethyl ether of plagiochin A (**97**) [90], and the same phenomenon has also been observed in marchantins and riccardins [30,31], as mentioned earlier. Plagiochins E (**7**) and F (**101**) were C7′-C8′ dihydroisoplagiochin A and 10-hydroxy C7′-C8′ dihydroisoplagiochin A, respectively [33]. Plagiochins E (**7**) and F (**101**) were synthesized through intramolecular Wittig reactions [91] and are supposed to exist in nature based on energetic and biosynthetic considerations [92], but the second one was never been isolated [91]. Isoplagiochins A and B (**102a**–**103a**) were first isolated from the liverwort *Plagiochila fruticosa* Mitt. [68]. Further fractionation of the crude extract of the same species resulted in the isolation of isoplagiochins C and D (**104**–**105**), and their structures have been elucidated [68]. Furthermore, isoplagiochin A was isolated from culture gametophytes of *Heteroscyphus planus* (Mitt.) Schiffner [69]. The hydrogenation of this compound gives a dihydro derivative. The ^1^H and ^13^C spectra of of isoplagiachin C resemble those of isoplagiochin A (**102a**), except for the signal patterns of a D ring; an additional phenolic hydroxyl group is at the D ring in place of the ethereal oxygen in isoplagiachin A. Isoplagiochin C (**104**) forms a tetraacetate, and the location of the hydroxyl group at C-13′ and the whole structure was determined using HMBC, HMQC and NOESY [68]. Isoplagiochin A (**102a**) and planusin A (**106**) were obtained from culture cells of *H. planus*, and their structures were determined using spectral analysis [69].

Isoplagiochins C and D (**104**–**105**) were obtained in an enantiomeric mixture at 85:15 and 48:52 ratios, in favor of the enantiomer possessing the *p*–configuration (*p*-64 and *p*-65) [21]. Three novel bisbibenzyls, namely isoplagiochoin F (**107**), dihydroisoplagiochin (**108**) and a chlorinated derivative monochlorinated isoplagiochin D (**109**), were isolated from a *Plagiochila* sp. [67].

2,12-Dichloroisoplagiochin D (**110**), 12,7′-dichloroisoplagiochin D (**111**) and 12, 10′-dichloroisoplagiochin C (**112**) were isolated from *Herbertus sakurii* of Herbertaceae along with isoplagiochin C (**104**) and D (**105**), and 2,12-dichloroisoplagiochin D (**110**) and 12-chloroisoplagiochion D (**113**) were isolated from *Mastigophora diclados* (Brid. ex F.Weber) Nees, suggesting the close chemical relation of these two genera [70].

#### 3.2.6. Halogenated Bisbibenzyls

Bazzanins A, B, C, D, E, F, G, H, I and J (**114**–**123**) are isoplagiochin derivatives with two biphenyl linkages and are substituted with 1–6 chlorine atoms, whereas bazzanin K (**124**) is a dichlorinated macrocycle consisting of a phenanthrene and a bibenzyl moiety connected with two biphenyl linkages, which are isolated from *Bazzania trilobata* (L.) Gray [74]. The linkages of the bibenzyl moiety are detected through HMBC and NOESY [74]. Bazzanins M-S (**125**–**131**) were obtained from *Lepidozia incurvata* [19].

#### 3.2.7. Ptychantols

Three novel macrocyclic bisbibenzyls, named ptychantols A, B and C (**132**–**134**), possess a *trans*-stilbene moiety isolated from *Ptychantus striatus*. Ptychantol B was the first reported macrocyclic bisbibenzyls possessing two biphenyl linkages between rings A and C and between rings B and D. The methylation of ptychantol A gave a dimethyl ether, whose IR spectrum indicated neither hydroxyl nor carbonyl absorption bands, indicating the presence of two ether oxygen in this compound. Hydrogenation gave a dihydroderivative, showing that this compound contains one olefinic group. The substitution pattern was suggested based on the ^1^H, ^13^C NMR, HMBC and NOESY spectra [73]. Another compound named dihyrdroptychantol A (i.e., 1′, 13′-dihydroxyisomarchantin, **135a**) was isolated from the natural source *A. angusta* [72], asproposed earlier via a computational method [92].

#### 3.2.8. Polymorphatins

This is a bisbibenzyl derivate with a novel skeleton named Polymorphatin A (**136**) [64] with a 1,4 disubstituted aromatic ring (ring A) and three 1,2,4-trisubstituted aromatic ring (rings B–D); in addition, the ^13^C NMR spectra shown δ (C) at 36.2, 37.3, 38.7 and 38.1 though the spectral discrepancies were reported in natural and synthetic materials [28].

#### 3.2.9. Glaucescenolide and Bisbibenzyl Glaucescenolides (GBB)

Glaucescenolide (**137**) is a compound with a sesquiterpenoid skeleton reported first from *S. glaucescens*, a plant source, earlier known only from marine mollusks and sponges [93]. Glaucescenolide shows an IR spectrum at 3378 cm^−1^ for hydroxyl absorption and a carbonyl absorption at 1740 cm^−1^, indicating the presence of a *α*, *β* –unsaturated *γ*-lactone. Scher et al. (2002) first proposed that furan, a natural product, can be an intermediate linking the biosynthetic pathways for glaucescenolide, GBB A and GBB B (**138**–**139**) (Figure 6) [49].

## 4. Structure–Activity Relationship of Bisbibenzyls and Bibenzyls

Bisbibenzyl compounds show a number of bioactivities like antibacterial [94], antiviral [47,95,96], antifungal [94,97,98], cytotoxic [18,23,42,99], antioxidant [18,19,100], neuroprotective [81], antivenomous [101], HAT treatment [102], muscle relaxing [103] and others. Bibenzyl and bisbibenzyl compounds involved in diverse types of bioactivities, as established through in vitro, in vivo and in silico studies, are the main contents of the discussion below.

The pusilatin and riccardin classes of compounds from liverwort were tested for their inhibitory activities against HIV-1 reverse transcriptase (RT), HIV-2 RT, a mutant RT, avian myeloblastosis virus RT, DNA polymerase *β* and RNA polymerase [53]. None of the viruses or enzymes were inhibited except DNA polymerase *β*, which was inhibited at IC_50_ ranging from 5 μM to 38 μM [53]. However, studies on the influenza virus revealed the efficacy of bryophyte phytoconstituents possessing a particular substructure. The RNA-dependent RNA polymerase of influenza virus has an acidic protein (PA) subunit with endonuclease activity, which carries out the decapping of host pre-mRNAs, resulting in the initiation of the viral transcription process [104]. Hence, the endonuclease activity has become a target for studies on anti-influenza activities. Iwai et al. (2011) screened 33 phytochemicals using assays, like PA endonuclease inhibition in vitro and anti-influenza A virus. Marchantins A (**1**), B (**2**) and E (**4**); plagiochin A (**97**); and perrottetin F (**55**), isolated from liverworts, were found to inhibit influenza PA endonuclease activity in vitro [47]. The authos found that the aforementioned compounds are effective because they have a 3,4–dihydroxyphenethyl substructure in them, apparently contributing to the inhibition of PA endonuclease, as was already reported [105] from a study of thalidomide derivatives containing the same substructure of 3,4-dihydroxyphenethyl. Among the effective compounds, the authors found marchantin E to inhibit the growth of both influenza A (H3N2 and H1N1) and B viruses, whereas marchantin A (**1**) and perrottetin F (**55**) were inhibitory to the growth of influenza B only. In addition, all five of the above compounds also reduce viral infectivity, with marchantin E (**4**) showing the highest activity, probably because of its fitting and chelating model of the inhibition of PA endonuclease activity revealed in a docking simulation study [47]; see Figure 7.

**Figure 7 plants-12-04173-f007:**
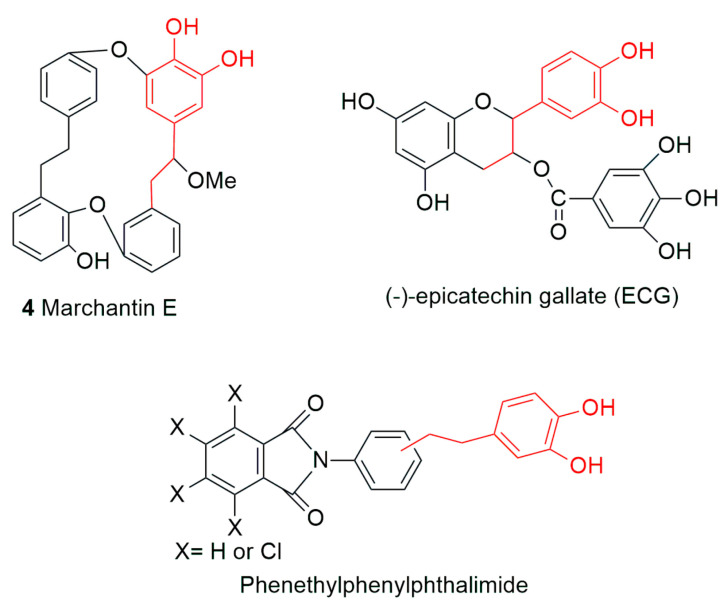
The structures of marchantin E (**4**), epicatechin gallate and phenethyphenylpthalimide adapted from [105].

The PA endonuclease activity of these compounds and other structurally similar compounds has been ascribed to the chelation of the manganese ions present in the active site of the enzyme through X-ray crystallography [106,107] and also interactions with different active site amino acids, as revealed by mutational studies [108]. The macrocyclic structure of marchantin-type compounds in combination with oseltamivir would be effective against the influenza virus.

Marchantin A (**1**) was first reported for its antibacterial activities [40,86], i.e., growth inhibition of several Gram-positive and Gram-negative bacteria with varying inhibition concentrations, and the findings were later corroborated [85], also the analogs and derivatives of this compound was tested for comparison. This macrocyclic compound possesses antibacterial activity, as it has diaryl ether or biphenyl bonds [109]. Marchantin A (**1**) was isolated from *M. emarginata* subsp. *tosana* (Stephani) Bischl., and it has been established that it induces cell growth inhibition in human MCF-7 breast cancer cells at IC_50_ 4.0 µg/mL. Marchantin A (**1**) induced a reduction in the cell viability of breast cancer cell lines A256, MCF7 and T47D (IC_50_ 5.5, 11.5 and 15.3 µM, respectively). The effect was dramatically increased in all cell lines in a synergistic manner. Marchantin A (**1**) possesses antimelanoma activity (IC_50_ 7.45–11.97 µg/mL) against human malignant melanoma cell lines (A375) [32,110,111].

4-Hydroxy-3′methoxybibenzyl (**140**) from a liverwort *Plagiochila stephensoniana* Mitt. exhibited antifungal activities due to the free hydroxyl group [112]. Two bibenzyls, viz., 3,4′-dimethoxyl-4-hydroxybibenzyl (**141**) and 3-hydroxy-4′-methoxybibenzyl (**142**), isolated from *Frullania muscicola* Stephani showed potent inhibitory effect against fungi [113]. Bisbibenzyls 6′,8′-dichloroisoplagiochin C (bazzanin B, **115**), isoplagiochin D (**105**) and 6′-chloroisoplagiochin D (bazzanin S, **131**) showed strong antifungal activities and large inhibition zones. Furthermore, bazzanin S (**131**) was compared with its permethylated derivatives against the pathogenic fungi, wherein bazzanin S showed a larger inhibition zone than the latter [75]. Asterelin A (**88**), asterelin B (**89**), 11-*O*-demethyl marchantin I (**143**), dihydroptychantol A (**135a**), marchantin H (**144**), marchantin M (**72**), marchantin P (**76**), perrottetin E (**13**), plagiochin E (**7**) and riccardin B (**12**) show moderate antifungal activity against common clinical pathogenic fungus *C. albicans*, with MIC value ranges from 16 μg/mL to 512 μg/mL. MIQ and MIC values of asterelin A are lower than asterelin B, indicating that the methylation of the OH groups decreases their antifungal activity [72]; see Figure 8.

**Figure 8 plants-12-04173-f008:**
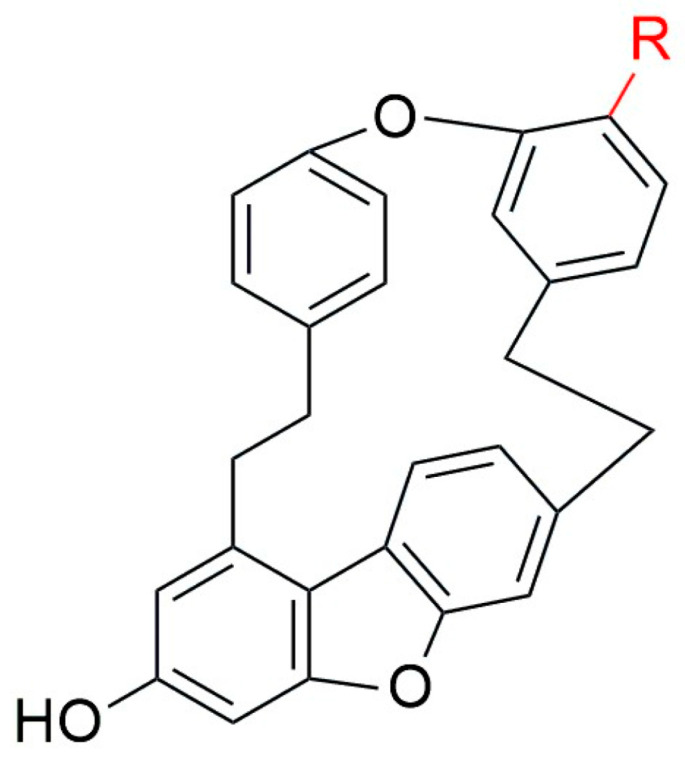
The structures of Asterelin A [R=OH] **88** and Asterelin B [R=OMe] **89** adapted from [72].

From the structure–activity point of view, previous studies also support the idea that the free hydroxyl group plays an important role in antifungal activity [33,75]. Riccardin D (**9**), another macrocyclic bisbibenzyl, inhibited biofilm formation by *C. albicans*, and it downregulated the hyphae-specific gene expression to inhibit hyphae formation [12]. Riccardin F (**79**) and riccardin C (**15**) showed antifungal activities against the fluconazole-sensitive and -resistant strains of *C. albicans*, with MIC values ranging from 32 μg/mL to 256 μg/mL, whereas pakyonol (**145**), neomarchantin A (**5**), isoriccardin C (**83**) and marchantin H (**144**) were inactive, wth MIC values >512 μg/mL. The presence of an additional aromatic *O*-methyl group may reduce the in vitro antifungal efficacy [60]. Macrocyclic bisbibenzyl may exert direct antifungal action by forming pores in the cell membranes, leading to fungal cell lysis [114]. Riccardin C (**15**), in combination with fluconazole with synergistic/additive activity, reduced the MIC from 256 μg/mL to <8 μg/mL. The possible reason for this drastic reduction seems to be the increased permeability of fluconazole and the inhibition of the drug efflux pump on the membrane [115]. Plagiochin E (**7**) acts as an antifungal agent and acts through the metacaspase-dependent apoptotic pathway in *C. albicans*. It induces G2/M cell cycle arrest, as evidenced by other cellular changes, contributing to the inhibition of cell cycle regulators CDC28, CLB2 and CLB 4 in *C. albicans* [116]. The diethyl ether extract of the liverwort *A. angusta* contained two minor bisbibenzyls, riccardin I (**82**) and angustatin A (**146**). Th latter one is the first non-heteroatom-containing variant possessing a bibenzylphenanthrene skeleton. Both variants possess moderate antifungal activities against the strains of fluconazole-sensitive and -resistant *C. albicans* [65].

Riccardin G (**81**), isolated from the Italian *Lunularia cruciata* (L.) Dumort. ex Lindb., showed cytotoxic activity against two human lung cancer cell lines, A549 and MRC5 [117]. Riccardin D (**9**) also demonstrates the induction of apoptosis against human non-small-cell lung cancer (NSCLCH460) and A549 (human non-small-cell lung carcinoma cells) and H460 (lung carcinoma epithelial cells) due to DNA topoisomerase-II inhibition [118].

The same bisbibenzyl (**9**) inhibited the proliferation of HUVECs (human umbilical vascular endothelial cells), decreased mortality and migration of HEVECs and vascular endothelial growth factor (VEGF) against human lung cancer H460 cell lines [119]. Riccardin D (**9**) also exhibited the prevention of intestinal adenoma (polyp) formation in APC^Min/+^ mice, a decrease in beta-catenin and cyclin D1 expression, the prevention of proliferation of intestinal polyps, and the triggeroing of apoptosis via the caspase-dependent pathway and te decrease in angiogenesis in intestinal polyps [120].

Riccardin D (**9**) may inhibit cell proliferation and induce apoptosis in HT-29 cells. This may be associated with the blocking of the NF-ΚB signaling pathway [121]. Hu et al., 2014, reported that the same bisbibenzyl (**9**) exhibited cytotoxic activity through the induction of apoptosis and the inhibition of angiogenesis and toposimerase-2. They also confirmed that apoptosis was not the sole mechanism by which riccardin D inhibits tumor cell growth, because a low concentration of riccardin caused cellular senescence in prostate cancer PCa cells, one of the most common malignant prostate tumors [122]. Riccardin D-26 derived from riccardin D (**9**) significantly inhibited cancer growth in both KB and KB/VCR xenografts without significant toxicity. It also inhibited cancer growth by inducing apoptosis in the activation of the mitochondria-mediated intrinsic apoptosis pathway. The above activity is stronger than that of the original riccardin D (**9**) [123].

Cyclic bisbibenzyls (marchantin A (**1**), B (**2**), D (**147**), E (**4**), riccardin C (**15**), isoriccardin C (**83**), acyclic bisbibenzyls (paleatin B, **95**), bibenzyls (perrottetin D, **10** and radulanin H, **14**) and NDGA were tested to see the inhibition efficacy in the key enzymes cyclooxygenase and 5-lipooxygenase 5-(LOX) of the arachidonic acid cascade. Acyclic paleatin B (**95**) showed the highest activity due to its best fit into the active site of the enzyme and is additionally stabilized by its lipophilic chain. Perrottetin D (**10**) also possessed the same activity, but it functions as a radical trap, existing as a free radical itself under the elevated temperature of the assay. Of marchantin A (**1**) and B (**2**), the latter showed two-fold more activity than the former, as it possesses two catechol moieties. Similarly, marchantin A (**1**) also showed inhibition on 5-lipooxygenase, as it possesses one catechol moiety in ring C and a phenolic hydroxyl group in ring A. Another compound of this series, i.e., marchantin D (**147**), reduced the activity five-fold due to the presence of a hydroxyethyl bridge, but the blocking of this hydroxyl group via methoxylation restores the activity almost to the level of marchantin A. Monophenols isoriccardin C (**83**), riccardin C (**15**) and radulanin H (**48**) exhibited the lowest reactivity among the compounds [124]; see Figure 9.

**Figure 9 plants-12-04173-f009:**
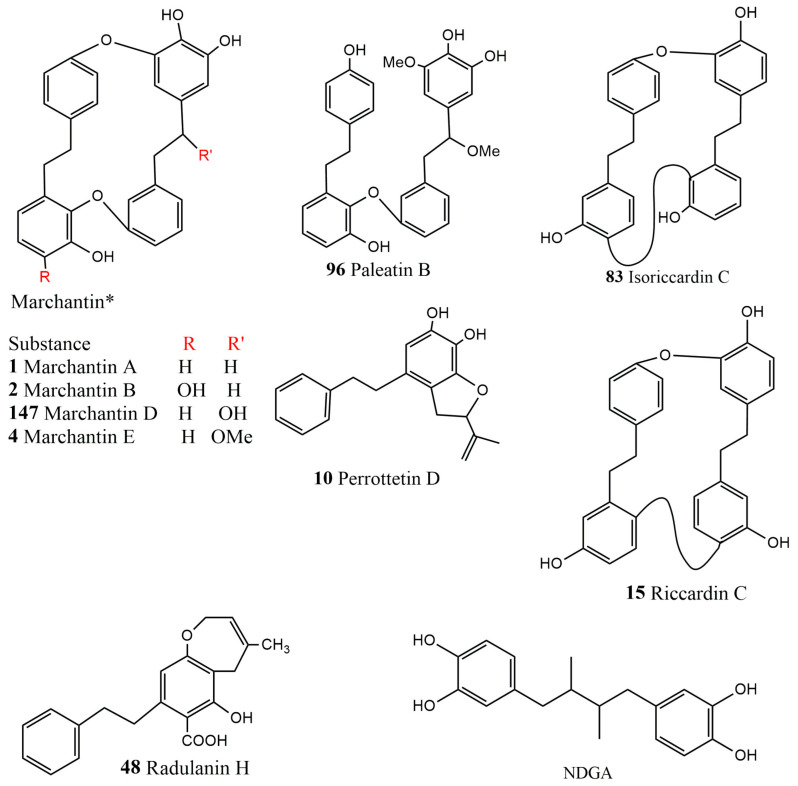
The structures of different bibenzyl and bisbibenzyls compared with another phenolic compound NGDA adapted from [124].

However, marchantin H (**144**) showed an IC_50_ value of 0.51 ± 0.03 μmol/L against the non-enzymatic iron-induced lipid peroxidation in rat brain homogenates. In the same study, marchantin H (**144**) also suppressed NADPH-dependent microsomal lipid peroxidation with an IC_50_ value of 0.32 ± 0.01 μmol/L without affecting the microsomal electron transport of NADPH-cytochrome P450 reductase. Marchantin H (**144**) can scavenge free radicals in the aqueous phase and act in a concentration-dependent manner. Marchantin H is a potentially effective and versatile antioxidant, and hence, it can be used as a chaperon protecting biomacromolecules against peroxidative damage [125].

Marchantin A trimethyl ether (**66**) was reported for having skeletal-muscle-relaxation properties compared to d-tubocurarine possessing the bisbibenzyl ether moiety, which differs from MATE in terms of the connection of the B and D rings only. d-TC combines with the nicotinic cholinergic receptor at the post-junctional membrane in the motor end-plate and thereby shows competitive inhibition with the transmitter action of acetylcholine. XRD data of MATE show the convex and concave surfaces surrounded by four benzene rings, and the central hole at the concave surface is responsible for the bioactivity [103]. Biological activity was explained with conformational analysis and computational chemistry, and the underlying mechanism of action was explained based on the minimum energy conformers [92,126]. Some bibenzyls with cyclic and acyclic ring structures were investigated for calmodulin inhibitory activities, and compounds like marchantin D and marchantin E showed a moderate effect due to the presence and nature of a nucleophilic atom at the ethano bridges, while the perrotettin E and perrotettin A showed increasing inhibitory concentrations due to the former being a linear bisbibenzyl and the latter being a monomeric stilbene where the macrocyclic ring influences the calmodulin-inhibitory activity [126]; see Figure 10.

**Figure 10 plants-12-04173-f010:**
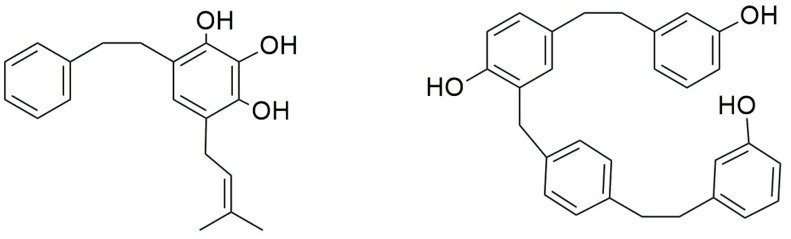
The structures of perrottetin A **52** and perrottetin E **13**.

Isoplagiochins A and B (**102a–103a**) show the inhibition of tubulin polymerization at IC_50_ 50 μM and 25 μM, respectively, but their dihydro derivatives (**102b**, **103b**) are less potent, with an IC_50_ > 100 μM, indicating that the restricted biaryl ring system of bisbibenzyls may be favorable for tubulin binding [127]; see Figure 11.

**Figure 11 plants-12-04173-f011:**
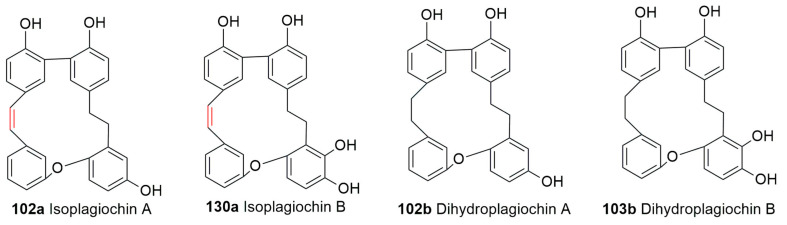
The structures of isoplagiochin A and B and their dihydroderivatives adapted from [127].

Genes regulating bile acid and cholesterol homeostasis are activated by the farnesoid X receptor (FXR), which has endogenous ligands, as well as some other non-steroidal and steroidal ligands. Suzuki et al. (2008) first reported marchantin A (**1**) and marchantin E (**4**), along with five bile acid derivatives that activated FXR in a reporter assay. The gene expression elevated by the screened compounds was much different for Cos-7, HepG2, HuH-7 and Caco-2 cells, and the genes regulated by FXR act in a cell-type-specific or gene-selective fashion, indicating the FXR target gene expression can be regulated by the molecular design of the compounds. Also, the screened compounds can be useful for studying FXR modulation, leading to selective FXR modulation for therapeutic use [48].

Glaucescenolide (**137**) is the most cytotoxic compound, with an IC_50_ of 2.3 µg/mL against P388 leukemia cells, and the mechanism of action shows similarity with other *α*, *β*-unsaturated sesquiterpene lactones, i.e., the alkylating activity, through the Michael addition of biological nucleophiles. Bisbibenzyl neomarchantin A (**5**) and B (**77**) showed weak to moderate cytotoxicity against P388 cells and antimicrobial activity against the Gram-positive bacterium *Bacillus subtilis* [49]. The inhibition potency of lipopolysaccharide-induced nitric oxide synthase (NOS) in RAW 264.7 macrophases was tested with marchantin A (**1**) and derivatives. Marchantin A (**1**) showed the strongest inhibition, with an IC_50_ of 1.44 μmol/L and the introduction of the hydroxyl group and methylation, and the positions of ether linkages in the ring regulate the inhibition [109]. Macrocyclic bisbibenzyls are also tested to see their role in apoptosis and cell cycle regulation, i.e., the major controlling mechanism of cancer cells. Marchantin A-treated MCF-7 cells show suppressed cyclin B1 gene expression, whereas marchantin C-treated human A172 glioma cells show increased cyclin B1 levels, indicating the different methods of cell cycle regulations of the two compounds [128]. Marchantin A (**1**) induces cytotoxicity and suppresses the proliferation of MCF-7 breast cancer cells via apoptosis through a caspase-dependent pathway and also influences the cell cycle [32]. Marchantin A (**1**) possesses two hydroxyl groups at C1′ and C6′ positions, which are very important parameters for scavenging free radicals like DPPH. The same two adjacent hydroxyl groups have been implicated in cytotoxicity induction, and, on the contrary, marchantin C has no hydroxyl group at the C6′ position, showing less antioxidant activity [128]. Later on, it was established that marchantin A and marchantin C both have the potential to develop candidate drugs for chemotherapy. Marchantin C induces the mitochondria-dependent intrinsic apoptosis pathway by arresting the G2/M phase of the cell cycle, as observed in vitro and in vivo. It upregulates Bax expression but downregulates Bcl-2 expression [128]. The study with marchantin C (**3**) was further extended by the same group of workers to see anti-invasiveness and antiangiogenic activity in glioma cells, and it was observed that marchantin C (**3**) inhibited the progression in a dose-dependent manner by affecting the MMP-2 activity via the MAPK pathway. However, further confirmation is required for a complete understanding of the mechanism of marchantin C (**3**) in brain cancer invasion and the migration of cancer cells [99].

A nitrogen-containing marchantin C and brominated products (**148–150**) prepared from marchantin C (**3**), as shown in Figure 12, possess cytotoxicity against KB, MCF-7 and PC3 (IC_50_ 6.3–27.2 µM). 10-Bromo-(**148**), 11-bromo-(**149**) and 12-bromomacrchantin C (**150**) are slightly more active than marchantin C (**3**). The dimer (**151**) and 12-*N*,*N*-dimethylaminomarchantin C (**152**) are less active than the brominated products [129]. Marchantin C (**3**) strongly inhibited the growth of human cervical tumor xenografts in a nude mouse model and decreased the quantity of microtubules in a time- and dose-dependent manner at the G2/M phase in human glioma tumor cells and HeLa (human cervical adenocarcinoma cell line) cells at 8–16 µM [99,128].

Zhang et al., 2015, reported that marchantin M (**72**) circumvented the growth of prostate cancer PC-3 tumors and upregulated expressions of CHOP and GRP78. The same authors indicated that marchantin M limited the proliferation and favored the apoptosis of DU145 cells in a time- and dose-dependent manner. Jian et al., 2013, found that marchantin M (**72**) induced autophagy-dependent cell death, which was accompanied by the induction of endoplasmic reticulum (ER) stress and the inhibition of proteasome activity in PCa cells [129,130].

**Figure 12 plants-12-04173-f012:**
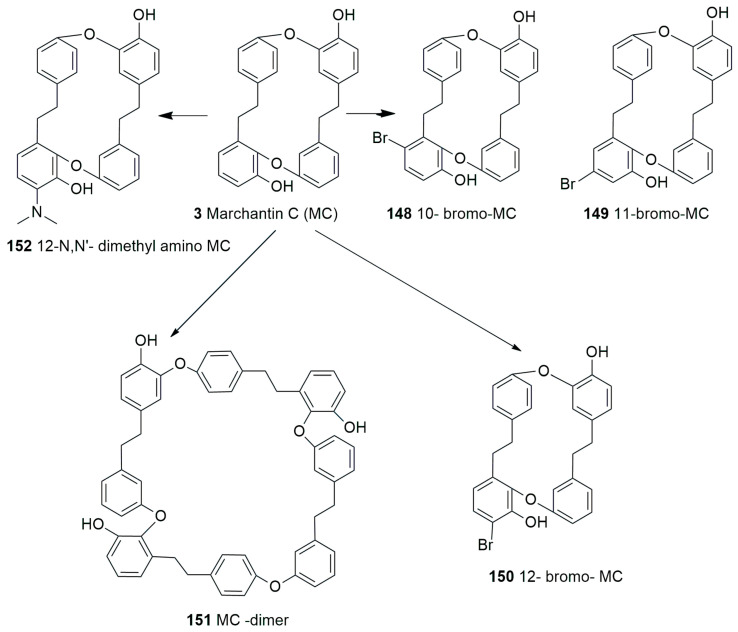
Preparation of bromomarchantin C (**148**–**150**) and dimeric marchantin C (**151**) from marchantin C (**3**).

Highly methoxylated bibenzyls, viz., 3,3′,4,4′-tetramethoxybibenzyl (**41**), chrysotobibenzyl (**42**), brittonin A (**39**) and brittonin B (**40**), were investigated to see the potentiality of inhibition in vincristine-resistant KB/VCR and adriamycin-resistant human myelogenous leukemia cells (K562)/A02 cells. Moderate cytotoxicity, with an LD_50_ value ranging from 11.3 μmol/L to 49.6 μmol/L, and the reversion of multi-drug resistance (MDR) with the reversal fold ranging from 3.19 μmol/L to 10.91 μmol/L at 5 μmol/L for vincristin resistant KB/VCR cells and 4.40 to 8.26 at 5 μmol/L for adriamycin-resistant K562/A02 cells were reported [55]. Plagiochin E (**7**) also has a reversal effect on MDR of K562/A02 cells [128]. DHA (**135a**), which are phenolic compounds, are the most promising MDR reversal agents. DHA and its structural analogs were tested on p-gp-mediated MDR in K562 and its multi-drug-resistant counterpart cells (K562/A02). DHA showed the largest potency to enhance adriamycin cytotoxicity in drug resistance cells with a reversal fold of 3.84–8.18, while its structural analogs were less effective. Phenolic groups binding to aromatic rings are necessary to increase the sensitivity of K562/A02 cells to adriamycin. When the phenolic groups are methylated, this activity is attenuated. On the other hand, the stilbenoid double bond in the structure has little effect on the reversal of MDR cells such as the hydrogenated structure in DHA. DHA (**135a**) acts as an antifungal product and has more potent MDR reversal activity by increasing the adriamycin cytotoxicity toward K562/A02 cells and vincristine cytotoxicity toward KB/VCR cells [131]. As mentioned above, DHA (**135a**) showed MDR reversal activity through the inhibition of the p-gp function and its expression, which prevents the efflux of drugs. However, some derived DHA compounds, i.e., with thiazole ring, such as the cyclic hexapeptide dendromide A, exhibited MDR reversal activity. Similarly, heterocyclic derivatives of combrestatins A, B, C and D, containing the bibenzyl moiety, were also cytotoxic to MDR cell lines [131]. In light of these results, researchers took an interest in the effect of thiazole moieties, and derivatives were synthesized to assess their biological activities, particularly MDR reversal activities, toward the cancer cell lines K56/A02 and KB/VCR and also molecular docking analyses to elucidate the binding modes of the analogs to p-gp [132]. DHA analogs were shown to be more potent in MDR reversal activities and increased vincristine cytotoxicity in KB/VCR cells, with the reversal fold ranges from 10.54 to 13.81 at 10 µmol/L, which is 3.2–4.3-fold stronger than DHA [132]. The exposure of U2OS cells to DHA treatment led to a remarkable growth inhibition in dose- and time-dependent manners, and the IC_50_ values of DHA for 24 and 48 hr to U2OS and U87 were 29.6, 24.7 and 21.2, 23.7 µM, respectively. DHA-induced autophagy is followed by apoptotic cell death accompanied by G2/M-phase cell cycle arrest in U2OS cells [133]. Pang et al., 2014, also evaluated dihydroptychantol A (**135a**) and its synthetic DHC derivative, DHA-2 (**135b**), against ovarian cancer cells. The exposure of ovarian cancer SKOv3 cell lines to DHA-2 resulted in the downregulation of the antiapoptotic X-linked inhibition of apoptosis protein (XIAP) and Bcl-2 and led to caspase-independent cell death [134]; see Figure 13.

**Figure 13 plants-12-04173-f013:**
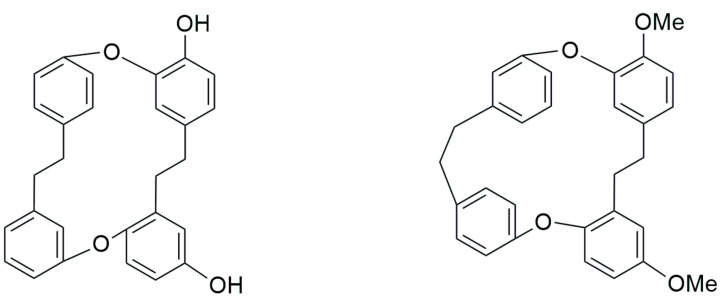
The structures of dihydroptychantol A **135a** and dihydroptychantol A2 **135b**.

Cytotoxicity against chemoresistant prostate cancer PC3 cells was treated with four cyclic bisbibenzyls, viz., riccardin C (**9**), pakyonol (**145**), marchantin M (**72**) and plagiochin E (**7**), and the underlying mechanism was observed to be the inhibition of proliferation and subsequent cell death through the induction of the apoptotic pathway via the downregulation of Bcl-2 expression and the upregulation of Bax expression by all four compounds, with IC_50_ values ranging from 3.22 μmol/L to 7.98 μmol/L. The effect of pakyonol was weaker than that of the other three compounds tested because pakyonol has one hydroxyl group in its structure and is methylated. The level of activity shown by the other three compounds at 10 μmol/L was achieved by pakyonol at only 20 μmol/L. Whether the -OH group and the adjacent oxygen bridge are essential features of the inhibitory effects of bisbibenzyls is not yet clear [135]. Forty-two selected bryophytes were screened in vitro to see the antiproliferative activity on human gynecological cancer cell lines (HeLa, A2780, and T47D), and potential antiproliferative activities were found [136].

A bibenzyl derivative 14-hydroxylunularin (**153**) was suggested for leishmanicidal therapy, as its bioactivity was established in in vitro conditions, with the activity depending upon the hybridization at the carbon–carbon bridge, the position, and a number of the free hydroxyl groups on the aromatic ring. In silico prediction using non-stochastic quadratic fingerprint-based algorithms also corroborated the in vitro results [137]; see Figure 14.

**Figure 14 plants-12-04173-f014:**
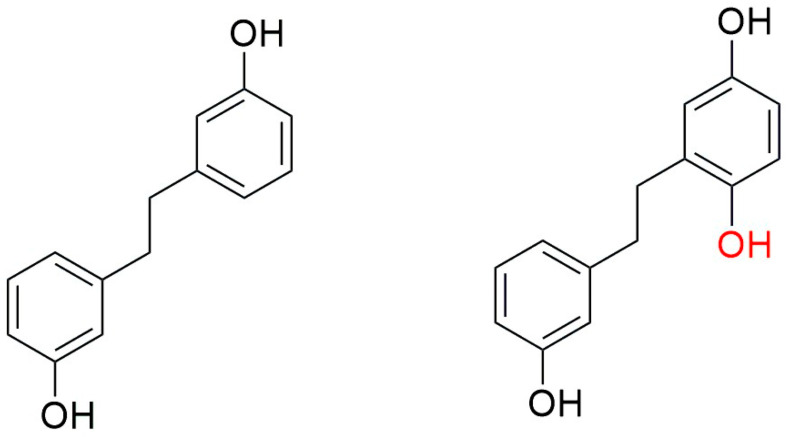
The structures of lunularin and 14-hydroxylunularin adapted from [137].

Almost all *Radula* species belonging to the Radulaceae are chemically very isolated from the other genera of liverworts, since they produce bibenzyls, prenyl bibenzyls and/or bisbibenzyls, along with their related compounds as major components, many of which possess cytotoxic effects against various cancer cell lines [79]. The Chinese *Radula apiculata* Sande Lac. ex Stephani elaborated radulapins A-H (**154**–**161**), which showed potent cytotoxic activity against PC-3, A549, MCF-7 and NCI-H12199 cancer cell lines at IC_50_ 1.4–9.8 µM [138].

Zhang et al., 2019, isolated nine prenyl bibenzyls from the Chinese *R. constricta* Stephani, six of which were racemate or scalemic mixtures, together with eleven related compounds. Among all of the prenyl bibenzyls, 2-carbomethoxy-3,5-dihydroxy-4(3-methyl-2-butenyl)-bibenzyl (**162**) indicated strong cytotoxicity against A549, NCI-H1299, HepG-2, HT-29 and KB at IC_50_ 6.0, 5.1, 6.3 and 6.7 µM, respectively. Cell death is triggered via mitochondria-derived paraptosis. Benzyl/o-cannabicyclol hybrid (1**63**) also had the same activity against only NCI-H1299 and A549 cancer cell lines at IC_50_ 7.5 and 9.8 µM, respectively [139].

The Chinese *R. sumatrana* Stephani elaborated four new pairs of bibenzyl/monoterpene hybrid enantiomers, named rasumatranins A-D (**164**–**167**), two new pairs of prenyl bibenzyl enantiomers, radulanins M (**168**) and N (**169**), known bibenzyl/o-cannabicyclol hybrid (**163**), 5-oxoradulanin A (**170**), radulanin A (**45**), 2-isopropenyl-6(β-phenylethyl)-4-hydroxy-2,3-dihydrobenzofuran (=tylimanthin B) (**171**) [18], radulanin I (**172**) and radulanin J (**173**), among which compound **170** showed the most potent cytotoxic activity against MCF-7, PC-3 and SMMC-7721 at IC_50_ 3.86, 6.60, and 3.58 µM. Compounds **164**, **165** and **172** also indicated the same moderate activity, as mentioned above [140].

Perrottetin E (**13**), 10′-hydroxyperrottetin E (**59**) and 10,10′-dihydroxyperrottetin E (**62**) showed modest activity on human leukemia cell lines HL-60, U-937 and K-562 and significant activity in human embryonal teratocarcinoma cell line (NT2/D1) and human glioblastoma cell line A-172 [141]. Twenty-eight compounds belonging to various classes, viz., phenol, quinolones, phenylpropanoid, terpenoid, and fatty acids, were detected via UHPLC-QTOF-MS metabolic profiling of the chloroform extract of *M. polymorpha*.

The main attributes, i.e., hepatoprotective activity, of these compounds are reported against hepatocellular injury caused by paracetamol-treated mice in dose-dependent manner, though preclinical studies are required to strengthen this conclusion [142].

The bibenzyls and bisbibenzyls discussed in the aforementioned section show a greater number of studies dealing with the screening of bioactivities and a lesser number of them focusing on the structure–activity relation. The moiety-based molecular docking study for future drug design is very low, which clearly shows that the last two phases of the clinical trial are required in a more structured way to commercialize these natural products for clinical uses.

## 5. Conclusions and Future Perspectives

Bryophytes, the minuscule creatures of nature, have great potential to produce bioactive secondary metabolites from which life-saving drugs can be obtained. The Chinese traditional medicine system has mentioned the use of about 50 bryophytes. As the primary colonizers of terrestrial ecosystem function, this group of plants is enriching and regulating the Earth’s biodiversity. Macrocyclic and acyclic bisbibenzyls quantified from different liverworts are now ongoing work platforms of several pharmacists and biochemists for establishing structure–activity relationships. Synthetic chemists are also interested in natural compounds with conformational chirality with no stereogenic carbon centers, as cavicularin-like unusual compounds are reported from natural products. In the forthcoming days, we hope to see a surge in liverwort-omics and the scaling up of the production of specific metabolites like bibenzyls and bisbibenzyls at the industrial level using recent knowledge of genome-editing and gene-expression tools.

## Figures and Tables

**Figure 1 plants-12-04173-f001:**
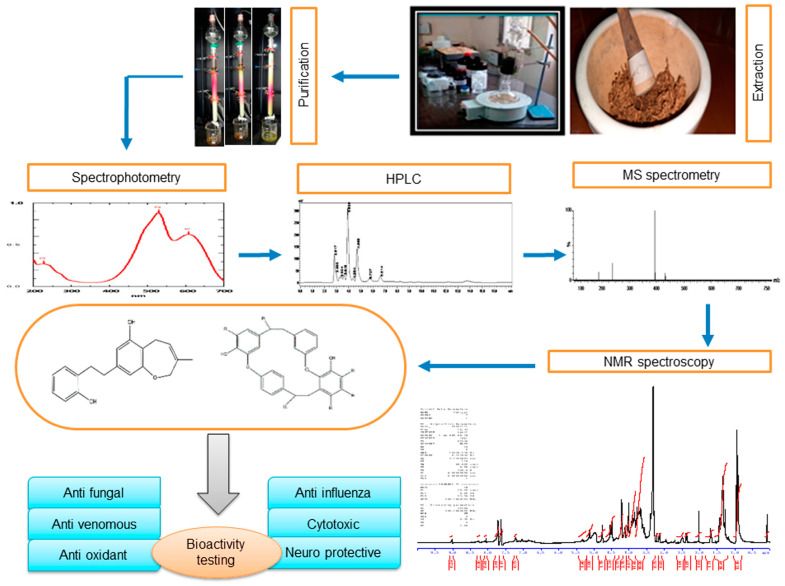
Schematic workflow of extraction, purification, characterization, structure elucidation and bioactivity testing.

**Figure 2 plants-12-04173-f002:**
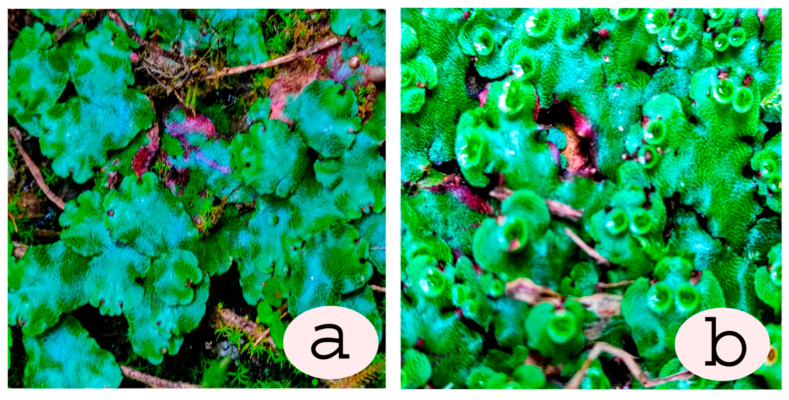
(**a**,**b**): Field photographs of *Marchantia* sp.

**Figure 3 plants-12-04173-f003:**
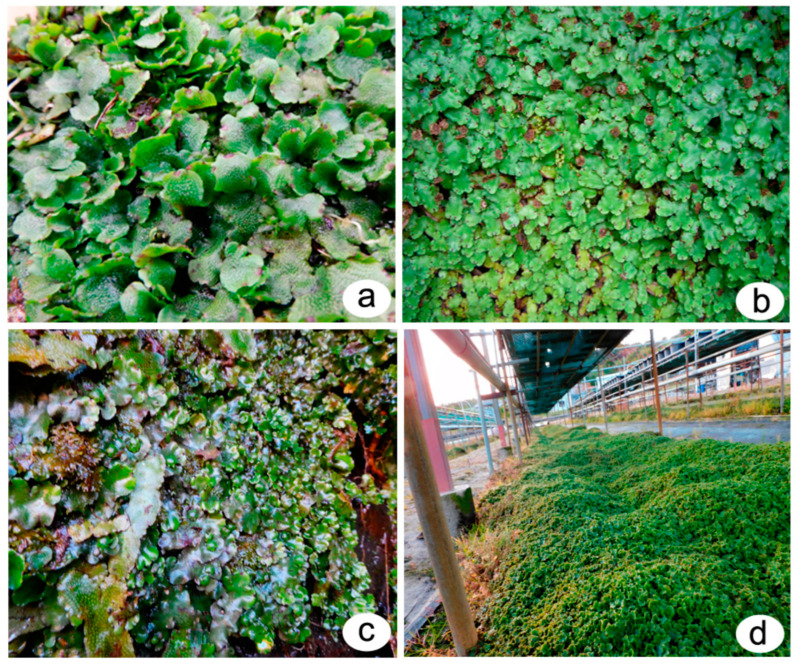
Photographs showing (**a**) *Conocephalum conicum*, (**b**) *Marchantia paleacea* subsp. *diptera*, (**c**) *Dumortiera hirsuta*, and (**d**) *M. polymorpha* growing in the research station (Tokushima Bunri University, Tokushima, Japan).

**Figure 4 plants-12-04173-f004:**
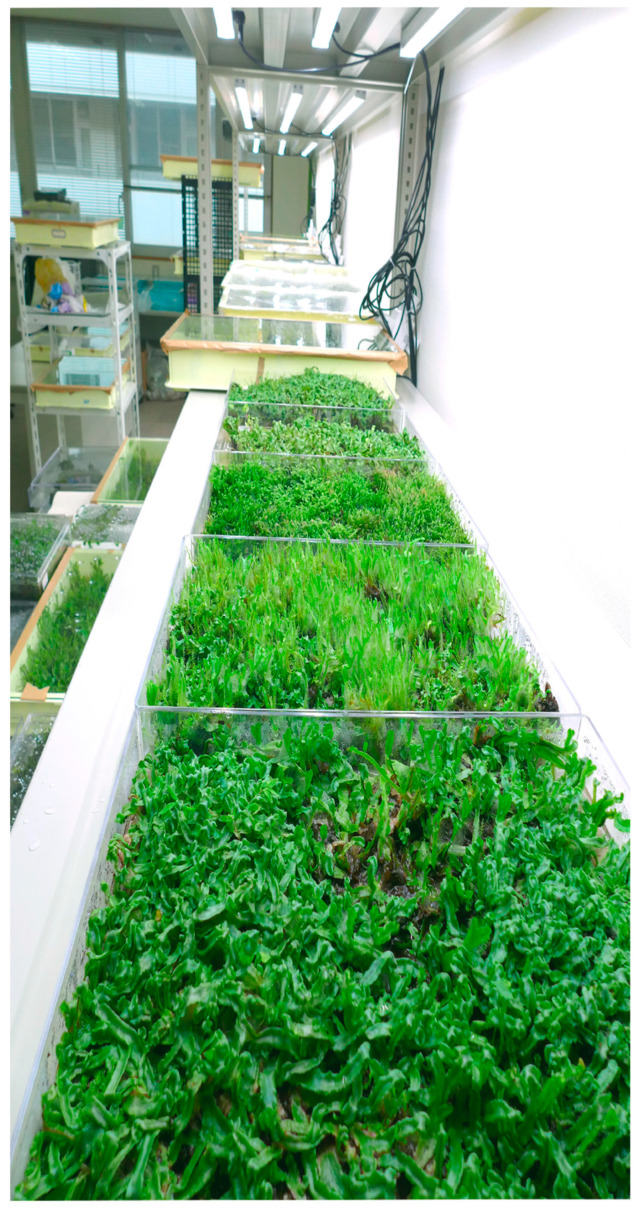
Culture room of liverworts showing *M. paleacea* subsp. *diptera* under LED light.

**Figure 5 plants-12-04173-f005:**
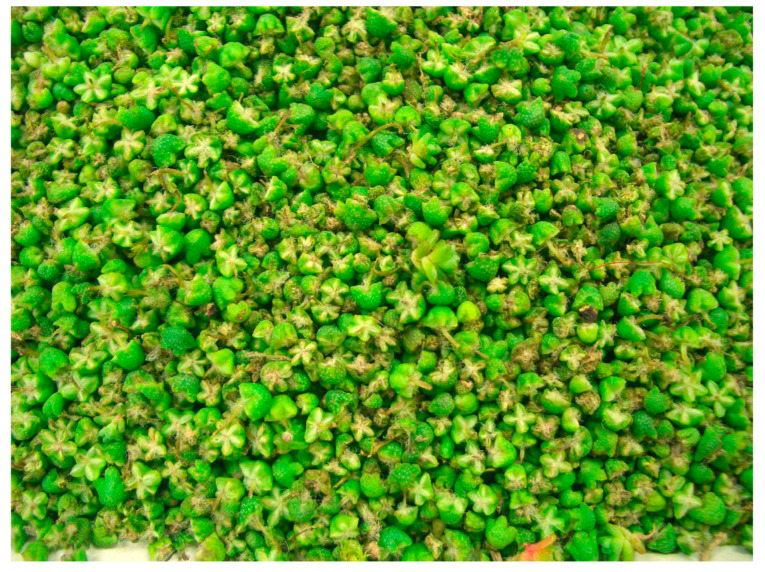
Photographs showing sporophytes of *Reboulia hemisphaerica* in the research station (Tokushima Bunri University, Tokushima, Japan).

**Figure 6 plants-12-04173-f006:**
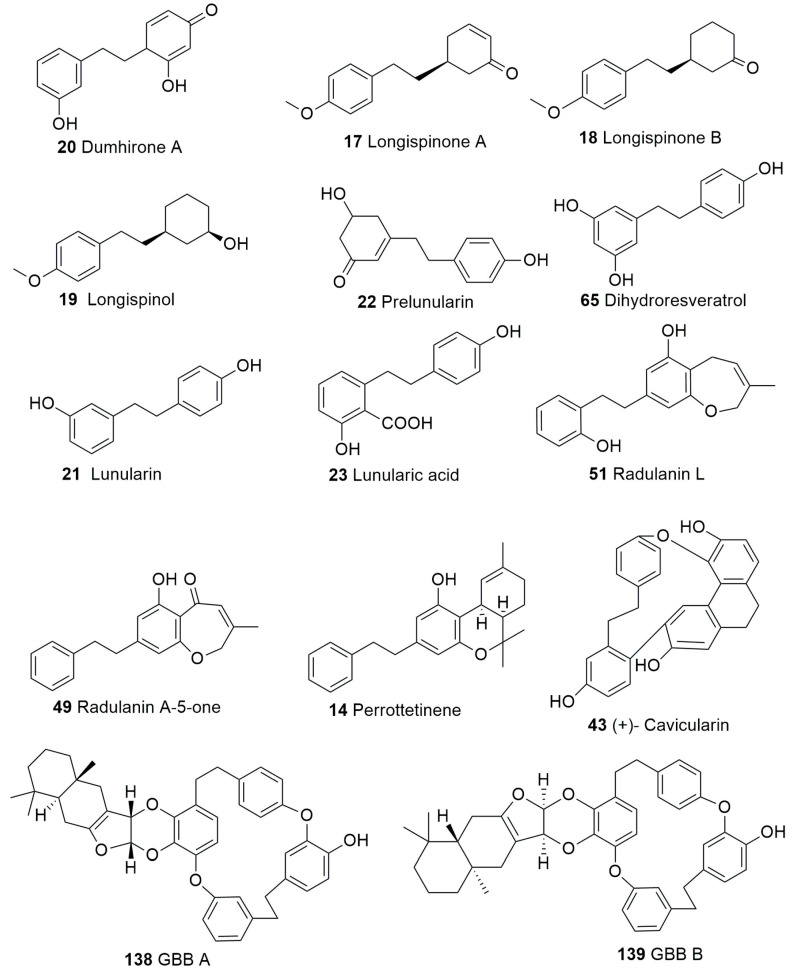
The structures of prebibenzyl, bisbibenzyl and bisbibenzyl–sesquiterpene ethers.

## Data Availability

Not applicable.

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
