# Peer review of "Recent Advances in the Phytochemistry of Bryophytes: Distribution, Structures and Biological Activity of Bibenzyl and Bisbibenzyl Compounds"

_plants, 2023, doi:10.3390/plants12244173_

Round 1
Reviewer 1 Report
Comments and Suggestions for Authors
This is a very useful contribution from the leader in the field and should be published. However, the manuscript is still in draft form and needs many things:
1. Send to an English proof reading service as definite articles, sense, tense etc are in need of editing.
2. Structures are missing some atoms.
3. Plant names are missing Latinisation in some cases and there are many places where a lack of spaces make it hard to read.
4. Finally, the references are appalling - in multiple styles and missing Latin names and some name errors for journals.
Once the above have been dealt with, the authors MUST roof read the final manuscript. I attach my annotations, but I gave up full correction - too many errors which could have been avoided.

See above
Author Response
|
Reviewer’s Comment |
Authors Reply (added with highlightes in revised text) |
|
Line 6 |
done |
|
Line 10 |
done |
|
Line 53 |
done |
|
Line 56 |
done |
|
Line 58 |
done |
|
Line 77 |
done |
|
Line121 |
done |
|
Line 123 |
done |
|
Line 132 |
done |
|
Table 1 (n.a.) |
Full stop omitted in the entire table |
|
Line 183 |
done |
|
Line 191 |
done |
|
Line 214 |
done |
|
Line224 |
done |
|
Line225 |
done |
|
Line 229 |
done |
|
Line 232 |
done |
|
Line 278 |
done |
|
Line 280 |
done |
|
Line 284 |
done |
|
Figure 6 |
Structure 65 and 139 corrected (earlier file no./ was Fig 7 i.e. corrected here) |
|
Line 326 |
done |
|
Line 333 |
done |
|
Line 340 |
done |
|
Line 352 |
done |
|
Line 387 |
done |
|
Line 390 |
done |
|
Line 401 |
done |
|
Line 404 |
done |
|
Line 510 |
done |
|
Line 567 |
done |
|
Line 621 |
done |
|
Fig 9 |
Structure 96 and 48 corrected (Fig. 10 has changed to Fig.9) |
|
Fig 10 |
Corrected (earlier fig 11) |
|
|
|
|
Fig 11 |
Corrected )earlier it was fig.12) |
|
Fig 14 |
Corrected earlier Fig.15 |
|
Line 817 |
done |

Reviewer 2 Report
Comments and Suggestions for Authors
Dear Authors,
The present study evaluates the advances in the phytochemistry of Bryophytes, with a focus on the biological activity of bibenzyls and bisbibenzyls. The research subject is interesting and brings scientific important data in the field. Some changes of the manuscript should nevertheless be performed in order to improve its quality. Following specific changes should thus be performed:
Minor changes
All scientific names of species should be italic. Please consult https://powo.science.kew.org/ or https://mpns.science.kew.org/mpns-portal/ for correct names of vegetal species. Moreover, please use correct nomination of species: first time they appear in text, you name them with full latin name and paternity and just afterwards you can use the abbreviation of the species.
I think that size of Table 1 can be reduced by editing the size of columns.
Major changes
Title: I do not understand the purpose of using the word “distribution” – does it refer to the compounds or to species of bryophytes?
Abstract: This section does not offer an overview of the study. It has way too many generalities and almost nothing specifically related to the study. The purpose of the study is not clear. This part should specifically state in 1-2 phrases each section of the manuscript.
Introduction: This section should offer an overview or a state of the art regarding the subject chosen for study. Only reduced informations are offered. Similar studies existing in scientific literature should be presented and authors need to compare the purposes of the study with these studies. Afterwards, authors need to clarify what the present study brings in novelty. It is very important to state what exactly you bring in novelty in order to express your originality. In fact, I cannot understand where you present generalities, where you state the purpose of the study and what is the purpose of the images. In fact, images are not even mentioned in text. Paragraphs are not separated. There is no proper connection between different concepts of the Introduction. The purpose of the study needs to be found in the last paragraph and be clearer. I do not undestand from this section if you refer to specific bryophytes (species) or you selected a family/clade that you treat specifically. Please modify accordingly the whole section. This section needs serious review.
I think that in your specific part you need to perform comparison with similar studies (some comparisons are performed) in term of findings, so that you can be able to emphasize novelty and originality of your study in this section once again, in terms of findings and not purposes, like in the Introduction.
A review should offer a cristical discussion of the subject, so you need to perform this, there are some sections that lack these discussions.
Conclusions: Please remove references, they should not be found in this section.
References and their editing do not follow the recommendations of the journal.
All these suggested changes should be performed in order to bring further improvements to the manuscript.
Comments on the Quality of English LanguageEnglish language needs moderate changes.
Author Response
|
Reviewer’s Comment |
Authors Reply (added with highlightes in revised text) |
|
All scientific names of species should be italic. Please consult https://powo.science.kew.org/ or https://mpns.science.kew.org/mpns-portal/ for correct names of vegetal species. Moreover, please use correct nomination of species: first time they appear in text, you name them with full latin name and paternity and just afterwards you can use the abbreviation of the species.
|
This point is covered e.g. Section2 Marchantia polymorpha L. (highlighted) |
|
I think that size of Table 1 can be reduced by editing the size of columns.
|
Reduced and fit to text |
|
Title: I do not understand the purpose of using the word “distribution” – does it refer to the compounds or to species of bryophytes |
“distribution” refers to compounds present in different species…for eg. Marchantin A first time reported from Marchantia polymorpha but later on from different species of the genus Marchantia or different liverwort taxa are compiled from the taxa these compounds are reported |
|
This section does not offer an overview of the study. It has way too many generalities and almost nothing specifically related to the study. The purpose of the study is not clear. This part should specifically state in 1-2 phrases each section of the manuscript |
Please see the present version it is highlighted where the purpose is written |
|
This section should offer an overview or a state of the art regarding the subject chosen for study. Only reduced informations are offered. Similar studies existing in scientific literature should be presented and authors need to compare the purposes of the study with these studies. Afterwards, authors need to clarify what the present study brings in novelty. It is very important to state what exactly you bring in novelty in order to express your originality. In fact, I cannot understand where you present generalities, where you state the purpose of the study and what is the purpose of the images. In fact, images are not even mentioned in text. Paragraphs are not separated. There is no proper connection between different concepts of the Introduction. The purpose of the study needs to be found in the last paragraph and be clearer. I do not undestand from this section if you refer to specific bryophytes (species) or you selected a family/clade that you treat specifically. Please modify accordingly the whole section. This section needs serious review |
This part is revised and purpose or objectives is mentioned in the last para i.e. highlighted. Here not a single genus or family is selected. As this review covers the distribution of bibenzyls and bisbibenzyls compound discovered till date since its first discovery, the available literatures from different database was covered.
Please see highlighted phrase |
|
Conclusions: Please remove references, they should not be found in this section.
|
Removed |
|
References and their editing do not follow the recommendations of the journal |
Edited in journal style. please see this version |

Round 2
Reviewer 1 Report
Comments and Suggestions for Authors
The revisions are now fine. Well done on a nice paper which will be well cited.
Author Response
As the file is ok..no further revision required
Reviewer 2 Report
Comments and Suggestions for Authors
Dear Authors,
The present study evaluates the advances in the phytochemistry of Bryophytes, with a focus on the biological activity of bibenzyls and bisbibenzyls. The authors performed some of the suggested changes after the first round of review. Following specific changes should still be performed:
Minor changes
Please do not forget to correct the names of species in legends of figures.
Major changes
Title: If the word “distribution” refers to the compounds, then add “of compounds”, because it is not clear.
Abstract: The purpose of the study is still not clear.
Introduction: This section is not significantly improved, so I will leave some of my my previous comments here. Similar studies existing in scientific literature are still not presented. Novelty and originality are not clear. In fact, I cannot understand where you present generalities, where you state the purpose of the study and what is the purpose of the images. Images are not even mentioned in text. There is no proper connection between different concepts of the Introduction. The purpose of the study needs to be found in the last paragraph and be clearer. I do not undestand from this section if you refer to specific bryophytes (species) or you selected a family/clade that you treat specifically. This section still needs serious review.
I still did not find the part where you perform comparison with similar studies in term of findings, so that you can be able to emphasize novelty and originality of your study in this section once again, in terms of findings and not purposes, like in the Introduction. This observation is not even addressed by authors.
Another observation that is not even addressed by authors is related to the cristical discussion of the subject, that are not offered and should be, according to the article type.
Overall, I find that the manuscript is not significantly improved after the first round of review.
All these suggested changes should be performed in order to bring further improvements to the manuscript.
Comments on the Quality of English LanguageEnglish language is fine, minor changes are required.
Author Response
.
|
Question of Reviewer |
Response of Author |
|
Minor changes Please do not forget to correct the names of species in legends of figures |
Revised |
|
Major changes Title: If the word “distribution” refers to the compounds, then add “of compounds”, because it is not clear |
Added and highlighted with blue font...the word compounds added |
|
Abstract: The purpose of the study is still not clear.
|
Rewritten and highlighted with blue font |
|
Introduction: This section is not significantly improved, so I will leave some of my my previous comments here. Similar studies existing in scientific literature are still not presented. Novelty and originality are not clear. In fact, I cannot understand where you present generalities, where you state the purpose of the study and what is the purpose of the images. Images are not even mentioned in text. There is no proper connection between different concepts of the Introduction. The purpose of the study needs to be found in the last paragraph and be clearer. I do not undestand from this section if you refer to specific bryophytes (species) or you selected a family/clade that you treat specifically. This section still needs serious review. I still did not find the part where you perform comparison with similar studies in term of findings, so that you can be able to emphasize novelty and originality of your study in this section once again, in terms of findings and not purposes, like in the Introduction. This observation is not even addressed by authors. Another observation that is not even addressed by authors is related to the cristical discussion of the subject, that are not offered and should be, according to the article type. Overall, I find that the manuscript is not significantly improved after the first round of review.
|
Images provide are renumbered and properly connected with text. Not a family or clade is reviewed here but the liverworts reported for bibenzyl and bisbibenzyl compounds are surveyed. Also other bryophytic taxa other than liverworts are reported with these bibenzyl and bisbibenzyl compounds are presented as reported in literature
The critical comparison and the limitations are highlighted with blue font and some major changes are done to maintain the flow. |
.

Round 3
Reviewer 2 Report
Comments and Suggestions for Authors
Dear Authors,
The present study evaluates the advances in the phytochemistry of Bryophytes, with a focus on the biological activity of bibenzyls and bisbibenzyls. The authors performed some of the suggested changes after the second round of review. Following specific changes should still be performed:
Major and minor changes
Names of species are not corrected in titles of figures.
Abstract: You have a repetition “Small phenolic molecules”. Line 25: which compound?
Introduction: I still find that there is no proper connection between different concepts of the Introduction. Maybe adding paragraphs and connecting them would make everything clearer.
All these suggested changes should be performed in order to bring further improvements to the manuscript.
Comments on the Quality of English LanguageEnglish language needs minor changes.
Author Response
The present study evaluates the advances in the phytochemistry of Bryophytes, with a focus on the biological activity of bibenzyls and bisbibenzyls. The authors performed some of the suggested changes after the second round of review. Following specific changes should still be performed:
Major and minor changes
Names of species are not corrected in titles of figures.
ANS: Corrected highlighted in blue font
Abstract: You have a repetition “Small phenolic molecules”. Line 25: which compound?
Ans ……….It means bisbibenzyls and bibenzyls see line 12
Introduction: I still find that there is no proper connection between different concepts of the Introduction. Maybe adding paragraphs and connecting them would make everything clearer.
Ans. Rewritten…please see blue font with yellow highlight part
All these suggested changes should be performed in order to bring further improvements to the manuscript.
